# A SPOPL/Cullin-3 ubiquitin ligase complex regulates endocytic trafficking by targeting EPS15 at endosomes

Michaela Gschweitl[1†], Anna Ulbricht[1†], Christopher A Barnes[1‡], Radoslav I Enchev[1], Ingrid Stoffel-Studer[1], Nathalie Meyer-Schaller[1§], Jatta Huotari[1¶], Yohei Yamauchi[2], Urs F Greber[2], Ari Helenius[1*†], Matthias Peter[1*†]

[1]Institute of Biochemistry, Department of Biology, Eidgenössische Technische Hochschule Zürich, Zurich, Switzerland; [2]Institute of Molecular Life Sciences, University of Zurich, Zurich, Switzerland

*For correspondence: ari. helenius@bc.biol.ethz.ch (AH); matthias.peter@bc.biol.ethz.ch (MP)

[†]These authors contributed equally to this work

Present address: [‡]Novo Nordisk Research Center, Seattle, Switzerland; [§]Department of Biomedicine, University of Basel, Basel, Switzerland; [¶]nspm ltd, Meggen, Switzerland

Competing interests: The authors declare that no competing interests exist.

**Abstract** Cullin-3 (CUL3)-based ubiquitin ligases regulate endosome maturation and trafficking of endocytic cargo to lysosomes in mammalian cells. Here, we report that these functions depend on SPOPL, a substrate-specific CUL3 adaptor. We find that SPOPL associates with endosomes and is required for both the formation of multivesicular bodies (MVBs) and the endocytic host cell entry of influenza A virus. In SPOPL-depleted cells, endosomes are enlarged and fail to acquire intraluminal vesicles (ILVs). We identify a critical substrate ubiquitinated by CUL3-SPOPL as EPS15, an endocytic adaptor that also associates with the ESCRT-0 complex members HRS and STAM on endosomes. Indeed, EPS15 is ubiquitinated in a SPOPL-dependent manner, and accumulates with HRS in cells lacking SPOPL. Together, our data indicates that a CUL3-SPOPL E3 ubiquitin ligase complex regulates endocytic trafficking and MVB formation by ubiquitinating and degrading EPS15 at endosomes, thereby influencing influenza A virus infection as well as degradation of EGFR and other EPS15 targets.

## Introduction

Endocytic trafficking is an essential cellular process for nutrition absorption, signal transduction, cell-cell communication and maintenance of cell homeostasis (*Doherty and McMahon, 2009*). Endocytosis is used for the uptake of exogenous and cellular cargo, including many viruses such as influenza A virus (IAV) (*Cossart and Helenius, 2014*; *Edinger et al., 2014*) or a cohort of plasma membrane proteins such as the epidermal growth factor receptor (EGFR) with their respective ligands (*Tomas et al., 2014*). Cargo entering the endosomal system can be delivered via early (EE) and late endosomes (LE) to lysosomes (LY) for degradation, or alternatively, recycled back to the plasma membrane (*Maxfield, 2014*; *Wandinger-Ness and Zerial, 2014*).

Transfer of cargo from EEs to LYs depends on an endosomal maturation process that involves a variety of protein- and lipid-based remodeling events. They include a small GTPase RAB5-to-RAB7 switch, a PtdIns(3)P to PtdIns(3,5)$P_2$ conversion, and changes in the luminal ion concentrations, most notably a decrease in pH (*Huotari and Helenius, 2011*). Together these changes prepare the endosomes for fusion with other LEs and LYs. Endosome maturation also involves sorting of membrane cargo destined for degradation into intraluminal vesicles (ILVs), thereby generating late endosomal vacuoles referred to as multivesicular bodies (MVBs) (*Piper and Katzmann, 2007*).

Ubiquitin has emerged as an important regulator of endocytosis and cargo degradation (*Clague et al., 2012*). It serves as a signal for membrane quality control, endocytic internalization,

**eLife digest** Individual cells can move material, collectively referred to as cargo, from the outside environment into the cell interior via a process known as endocytosis. The cell then has different routes to transport the packages of cargo, called endocytic vesicles, to specific locations within the cell. Protein-based molecular machines move the cargo and control how it is selected and targeted to different destinations. For example, a molecular machine that contains a protein called CUL3 labels other components of the system with a chemical tag to regulate the route cargo takes in mammalian cells. However, it was not clear how CUL3 can selectively attach the chemical labels.

Gschweitl, Ulbricht et al. have now found that another protein called SPOPL provides selectivity for the CUL3-based machine during endocytosis in human cells. The experiments show that SPOPL attaches to endocytic vesicles, and that CUL3 and SPOPL work together to label a specific component of these vesicles called EPS15. The label changes how EPS15 interacts with other proteins. When SPOPL is not present in a cell, EPS15 is unnaturally stable and occupies many of the routes used by endocytic cargos. The cargo directly interacting with EPS15 is then routed on the fast lane to its destination, while other cargo accumulate in a kind of molecular traffic jam.

Other proteins like SPOPL are specific for the endocytic system. Exchange of SPOPL with these similar proteins in the CUL3 machine is likely to chemically label a different set of endocytic proteins. Gschweitl, Ulbricht et al.'s next challenge is to identify the selectivity, targeting and coordination of these exchangeable components in the endocytic system.

endosome maturation, and cargo sorting into ILVs by the endosomal sorting complexes required for transport (ESCRTs) (*Piper et al., 2014*). Indeed, ubiquitination is indispensable for the interaction of EGFR with the ESCRT-0 complex member HRS and thus for efficient lysosomal targeting (*Eden et al., 2012*). Specific deubiquitination enzymes (DUBs) like AMSH and UBPY associate with ESCRT components to cleave and recycle ubiquitin (*Alwan and van Leeuwen, 2007*; *Meijer et al., 2012*). Ubiquitination also regulates the machinery involved in molecular sorting and the ILV formation, although this regulatory layer is less understood.

In a recent study, we observed that Cullin-3 (CUL3) is involved in regulating endosome maturation and endo-lysosomal trafficking in mammalian cells. In CUL3-depleted cells, late endosomes fail to mature properly. They are enlarged and often devoid of ILVs. EGFR degradation is delayed and its ligand EGF accumulates in LE/LYs. Moreover, the cells are resistant to influenza A virus (IAV) infection because instead of delivering their capsid components to the cytosol and the nucleus, the endocytosed virus particles remain associated with immature LE vacuoles (*Huotari et al., 2012*).

CUL3 is the scaffolding subunit common for a large subfamily of Cullin-RING E3 ubiquitin ligases (CRLs) that mainly use the RING-H2 finger protein RBX1 to recruit charged ubiquitin conjugating enzymes (E2s) and catalyse the transfer of ubiquitin onto substrates (*Lydeard et al., 2013*). This is stimulated by the conjugation of the ubiquitin-like protein NEDD8 to the cullin subunit (*Duda et al., 2008*; *Saha and Deshaies, 2008*) through an enzymatic cascade analogous to ubiquitination (*Enchev et al., 2015*). CUL3 may be activated at endosomes by DCNL3, an E3 ligase for NEDD8, known to associate with membranes by virtue of a covalent lipid modification (*Meyer-Schaller et al., 2009*). CRL activity is further modulated by deneddylation as well as non-catalytically by the COP9 signalosome (*Emberley et al., 2012*; *Enchev et al., 2012*; *Fischer et al., 2011*).

In addition to CUL3, RBX1, and NEDD8, the CRL3 complexes contain a subunit that binds to CUL3 through a Bric-a-brac/Tramtrack/Broad (BTB) domain. The BTB domain-containing proteins possess diverse substrate recognition domains and function as CRL3-specific substrate adaptors (*Pintard et al., 2004*). There are about 150 different BTB domain-containing proteins in the human proteome (*Stogios et al., 2005*), indicating that a large cohort of CRL3 substrates exists, most of which are presently unidentified.

To gain mechanistic insight into the role of CRL3 in endocytic trafficking, we identified relevant BTB-adaptors by siRNA screening. We found that cells depleted of the BTB-adaptor Speckle-type POZ protein-like (SPOPL) were defective for IAV infection and showed aberrant late endosomal vacuoles resembling those observed after CUL3 depletion. Although SPOPL shares 81% sequence

identity with the tumor driver Speckle-type POZ protein (SPOP) and these two adaptors dimerize in vitro (*Errington et al., 2012*), only SPOPL was found to associate with endosomes. We identified the endocytic adaptor EPS15 as a critical substrate of the CRL3$^{SPOPL}$ complex. Our results indicated that EPS15 ubiquitination by CRL3$^{SPOPL}$ is needed for efficient intraluminal vesicle formation during endosome maturation as well as for uncoating of IAV capsids.

## Results

### Influenza A virus infection and uncoating are dependent on the CRL3 substrate adaptor SPOPL

To identify the BTB-domain containing CRL3 substrate-adaptors involved in endosome maturation, we took advantage of the observation that IAV depends on CUL3 activity for cell entry and infection (*Huotari et al., 2012*). We performed siRNA screening in the lung adenocarcinoma cell line A549 depleted for 130 human BTB-containing proteins using expression of the viral nuclear protein (NP) as read-out after IAV addition (*Figure 1A* and *Supplementary file 1*). Depletion of 14 BTB proteins caused 50% or greater decrease in the number of IAV infected cells. Previously described assays (*Banerjee et al., 2013*) were used to determine which step in the infection pathway was inhibited, including virus binding to cells, endocytic uptake, acid-conversion of viral hemagglutinin (HA), penetration by fusion, uncoating of the viral capsid, and nuclear import of the viral ribonucleoproteins (vRNPs). Out of 14 primary hits, we decided to follow up the BTB proteins SPOP and SPOPL because their depletion gave a phenotype of defective virus uncoating similar to that observed after CUL3 depletion. The other BTB-proteins identified in this screen were required for different steps of efficient IAV infection, and will be studied elsewhere.

To focus on the involvement of SPOP and it close homologue SPOPL in IAV infection, we used three siRNA oligonucleotides (siSPOP1-3 and siSPOPL1-3). They reduced the respective target gene expression efficiently as judged by quantitative RT-PCR. Notably, SPOP depletion also influenced mRNA levels of SPOPL, but not vice versa. Immunoblotting using a home-made, affinity-purified antibody specific for SPOPL confirmed downregulation of SPOPL at the protein level (*Figure 1—figure supplement 1A*). Knockdown of SPOP or SPOPL with the siRNAs resulted in a reduction of viral NP expression in A549 and HeLa cells to levels comparable to CUL3 depletion (*Figure 1B* and *Figure 1—figure supplement 1B*), but not as strong as upon depletion of a vATPase subunit (siATP6V1B2). As expected, cells that were depleted of either SPOP or SPOPL showed a block in IAV uncoating (*Figure 1F*, *Figure 1—figure supplement 1G*). As a consequence, subsequent nuclear import of NP was also impaired (*Figure 1G*, *Figure 1—figure supplement 1H*). Aside from virus binding to the cell surface, which was elevated in SPOPL-depleted cells, we found that endocytosis, acidification and fusion of the viral particles were comparable to controls (*Figure 1C–E*, *Figure 1—figure supplement 1C–F*).

The defects in IAV uncoating and nuclear import of vRNPs in SPOP or SPOPL-depleted cells were rescued by a brief acidic incubation of virus bound to the cell surface (*Figure 1H*). This is known to induce direct fusion of the viral particle at the plasma membrane, thus bypassing to some extent the need for endocytic trafficking (*Stauffer et al., 2014*). Together, these results indicated that SPOP or SPOPL depletion leads to a virus entry defect caused by inhibition of uncoating and release of viral capsid components from endocytic vacuoles.

### SPOPL, but not SPOP, associates with endosomes and regulates late endosome maturation

To further characterize the functions of SPOP and SPOPL, we used fluorescence microscopy and cell fractionation to determine their subcellular localization. We found that while GFP-tagged SPOP was predominantly nuclear when expressed in HeLa cells, GFP-tagged SPOPL was cytoplasmic and confined to puncta (*Figure 2—figure supplement 1A*). Endogenous SPOPL, contrary to SPOP, was mainly in the soluble cytosolic fraction after sedimentation of total cell extracts prepared from HeLa cells (*Figure 2A*). Moreover, endosome purification revealed that some of the SPOPL and CUL3 cofractionated with endosomal markers such as EPS15, HRS and STAM (*Figure 2B*). We concluded that of the two BTBs, SPOPL was more likely to directly influence endocytic trafficking due to its endosomal localization.

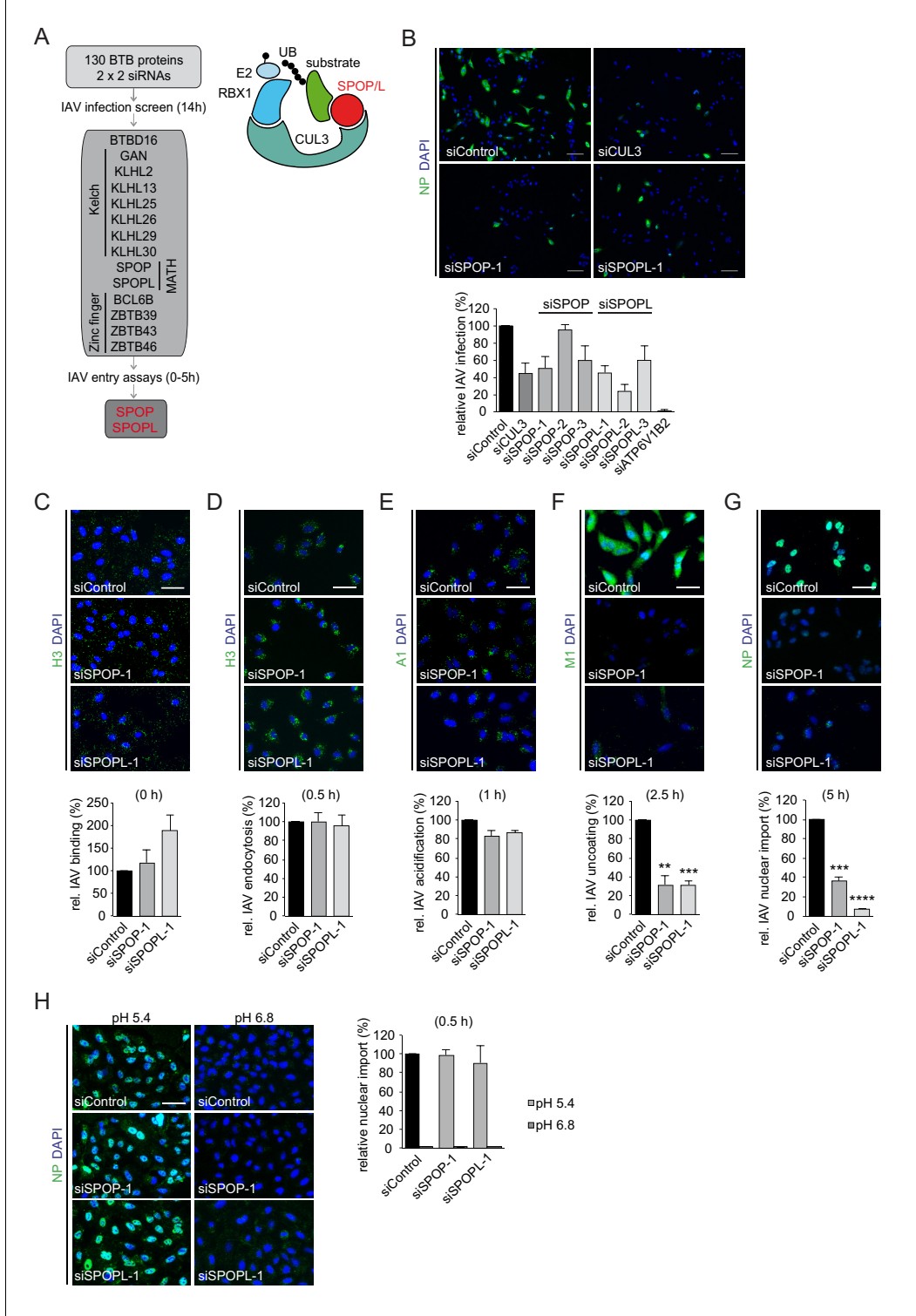

**Figure 1.** The CRL3 substrate adaptors SPOP and SPOPL are crucial for influenza A virus (IAV) infection and uncoating. (**A**) siRNA screen workflow for BTB adaptor proteins with similar IAV infection phenotypes as CUL3 (left). Schematic representation of the CRL3[SPOP/L] E3 ubiquitin ligase complex (right). CUL3 mediates the formation of ubiquitin chains (UB) to its substrates by binding to the ubiquitin charged conjugating enzyme (E2-UB) via RBX1 on one side while allowing the interaction with the substrate through the substrate adaptor proteins SPOP or SPOPL on the other side. (**B**) IAV X31 infection assay. Images show A549 cells treated with control siRNA (siControl) or siRNA-depleted of CUL3 (siCUL3), and the BTB-adaptor SPOP (siSPOP-1) or SPOPL (siSPOPL-1) for

*Figure 1 continued on next page*

*Figure 1 continued*

72 hr before infection with IAV X31. IAV infection was quantified by co-staining the cells with NP specific antibodies and DAPI to indicate nuclei. Cells siRNA-depleted for the vATPase subunit ATP6V1B2 (siATP6V1B2) were included for positive control. Scale bar = 100 µm; Data are mean + SD, n > 100 cells per sample, N = 4. (C-G) IAV entry assays. A549 cells were treated with control, SPOP- or SPOPL-specific siRNA, and binding of IAV X31 to the cells was monitored by immunofluorescence staining of the hemagglutinin (HA) with anti-H3 antibody (C). The IAV infection was allowed for 0.5 hr to follow IAV endocytosis with HA staining (D), for 1 hr to monitor IAV acidification using A1 antibodies (E), for 2.5 hr to check IAV uncoating by M1 detection (F) and finally for 5 hr to track nuclear import of IAV vRNPs by NP-specific antibodies (G). Nuclei were stained with DAPI and entry steps quantified relative to control. Scale bar = 50 µm; Data are mean + SD, n > 500 cells per sample, N = 3. **$p \leq 0.01$, ***$p \leq 0.001$; ****$p \leq 0.0001$. (H) Acid-induced endocytic-bypass entry assay. IAV nuclear import after acid-induced fusion at the PM was monitored in A549 cells using indirect immunofluorescence staining for NP and counterstaining with DAPI for infection quantitation. Note that pH 5.4 allows acid-induced endocytic-bypass infection of IAV. Scale bar = 50 µm; Data are mean + SD, n > 500 cells per sample, N = 3.

The following figure supplement is available for figure 1:

**Figure supplement 1.** The CRL3 substrate adaptors SPOP and SPOPL are crucial for influenza A virus (IAV) infection and uncoating.

We next used immunoblotting, fluorescence microscopy and thin sectioning electron microscopy (EM) to examine the potential role of SPOPL in the endocytic system. When visualized by the marker GFP-RAB5 using live cell imaging, early endosomes appeared normal in SPOPL-depleted cells compared to control siRNAs (*Figure 2C*). In contrast, the late endosomal system as monitored by GFP-RAB7 was severely distorted when SPOPL was depleted by a siRNA or a shRNA construct (*Figure 2C* and *Figure 2—figure supplement 1B*). The diameter of GFP-RAB7 positive endosomes increased two fold and the swollen vacuoles were clustered in the perinuclear region. A similar defect was observed when CRL activity was pharmacologically inhibited by the addition of MLN-4924 (*Figure 2C*), which prevents cullin neddylation (*Soucy et al., 2009*). Moreover, the endosome-to-Golgi transport marker GFP-RAB9 revealed strongly enlarged vacuoles upon SPOPL depletion as well, implying a broad effect of SPOPL on the late endosomal transport system (*Figure 2—figure supplement 1C*). Qualitative EM analysis of SPOPL-depleted cells showed that endosomal vacuoles were empty or contained little dense material and few ILVs were detectable compared to control cells (*Figure 2D*). The formation of normal MVBs was thus abrogated.

To narrow down the role of SPOPL we analyzed the level of early and late endosomal markers as well as markers of recycling endosomes, ER, autophagy and cytosol (*Figure 2E*). In contrast to SPOP, depletion of SPOPL caused changes in late endocytic markers, in particular the MVB markers EPS15, HRS, STAM and TSG101. The level of most early endocytic and recycling markers did not change (Clathrin, EPSIN1, Caveolin, Calnexin, RAB11). Notable, out of the tested receptors, EGFR and MET revealed significant level changes, while levels of VEGFR, IGF1R and HER2 - a close relative of EGFR - were not altered.

We purified endosomes from whole cell lysates and further fractionated them by density gradient centrifugation. Analysis of the endosome fraction revealed that in contrast to the ESCRT components HRS and STAM and the endocytic adaptor EPS15, the lysosomal marker LAMP1 and early endocytic marker EEA1 were reduced in endosomal fractions upon SPOPL depletion (*Figure 2—figure supplement 1D*).

Considering both total protein levels and protein levels in the endosome fractions as well as microscopy data, we conclude that the late endosome system, most likely the ILV/MVB formation process, is regulated by SPOPL.

## EPS15 is a CRL3[SPOPL] ubiquitination substrate and is degraded via the 26S proteasome

Our results indicated that especially EPS15, but also ESCRT subunits HRS and STAM, accumulate in SPOPL-depleted cells. EPS15 is a scaffolding adaptor protein for clathrin-coated vesicles and a critical regulator of EGFR endocytosis and endosomal sorting (*Gucwa and Brown, 2014*; *Li et al., 2014*; *Roxrud et al., 2008*; *van Bergen en Henegouwen, 2009*). Interestingly, we found that EGFR

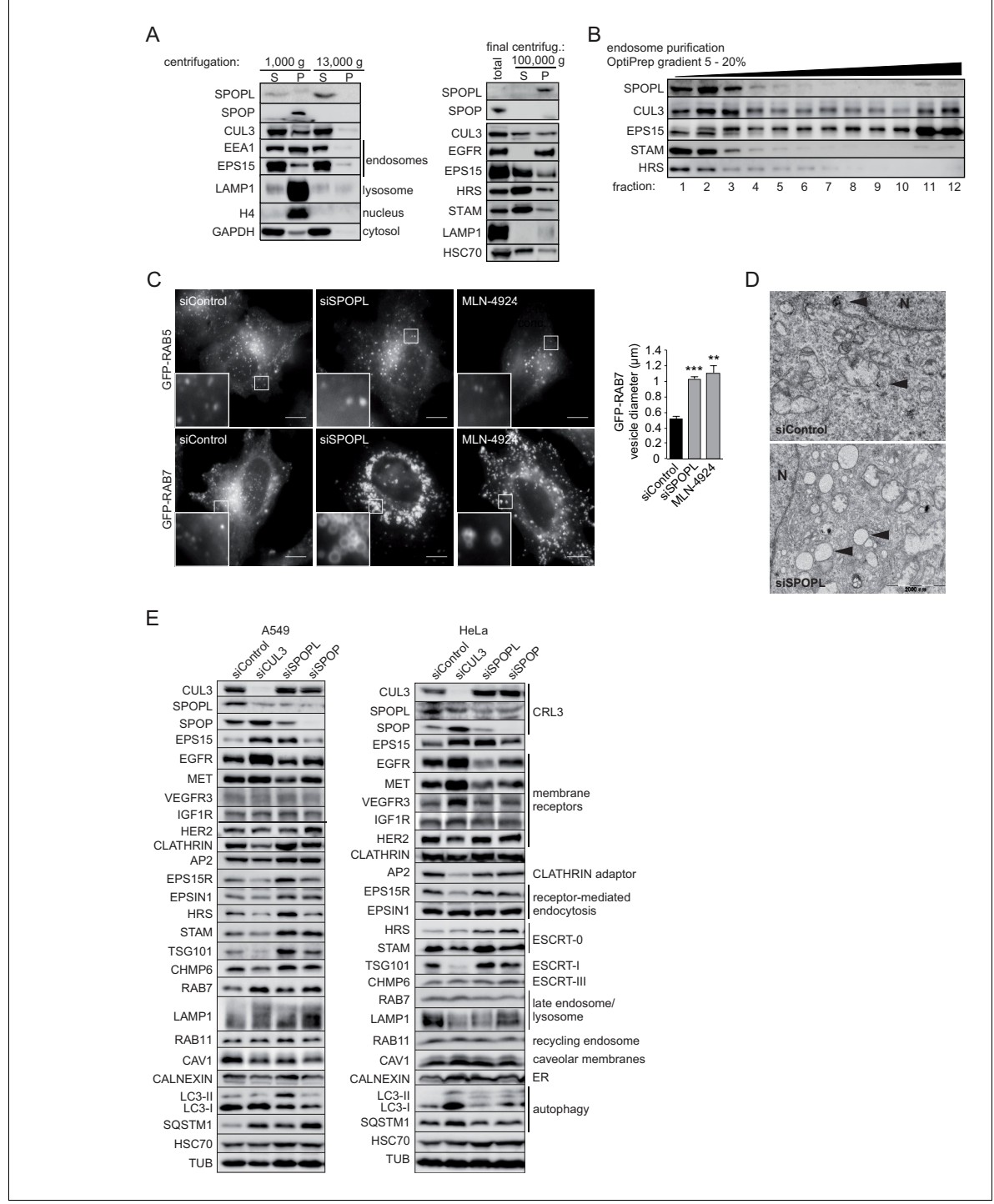

**Figure 2.** The BTB adaptor SPOP is nuclear, while SPOPL localizes to endosomes and affects endosome maturation. (**A**) Extracts prepared from HeLa cells were analyzed by differential centrifugation. The indicated proteins were probed in the supernatant (S) and pellet (P) fractions by immunoblotting after sedimentation of nuclei at 1000 g and after further fractionation of the resulting supernatant at 13,000 or 100,000 g. The cellular organelles detected by the specific antibodies are marked. (**B**) Endosomal organelles were enriched from HeLa cells by differential centrifugation and then fractionated on a 5 – 20% OptiPrep gradient. After centrifugation, TCA-precipitated gradient fractions (1–12) were analyzed by immunoblotting with specific antibodies against SPOPL, CUL3, EPS15, STAM and HRS. (**C**) HeLa cells expressing GFP-RAB5 and GFP-RAB7 were treated with siControl or siSPOPL and live cell imaging monitored their expression. A set of untransfected cells was treated with 10 μM MLN-4924 to inhibit CRL activity by preventing neddylation. Scale bar = 10 μm. Regions of interest (squares) are shown at 5x magnification. Vesicle diameter was quantified by Image J. Data are mean + SD, n = 100 vesicles per sample, N = 3. ***p≤0.001; **p≤0.01. (**D**) HeLa cells were depleted of SPOPL or using siControl, fixed after 72 hr and thin sections analyzed by electron microscopy (EM). MVBs are indicated with a black arrow head. Note the enlarged vacuoles in SPOPL-

*Figure 2 continued on next page*

*Figure 2 continued*

depleted cells that were found empty, devoid of ILVs. N: nucleus. Scale bar = 2000 nm. (**E**) A549 and HeLa cells were depleted of CUL3, SPOPL or SPOP for 72 hr, and then cell lysates were prepared and analyzed via immunoblotting with specific antibodies for markers of different cellular compartments. (CRL3 – Cullin-RING ligase 3, ER – endoplasmic reticulum)

The following figure supplement is available for figure 2:

**Figure supplement 1.** The BTB adaptor SPOP is nuclear, while SPOPL localizes to endosomes and affects endosome maturation.

was strongly reduced in SPOPL-depleted cells (*Figure 3A*), while it was elevated in cells depleted of CUL3 (*Huotari et al., 2012*) and in cells treated with the CRL-inhibitor MLN-4924 (*Figure 3—figure supplement 1A*). Conversely, overexpression of HA-tagged SPOPL caused a two fold increase in EGFR and a 50% decrease in EPS15 (*Figure 3B*). EPS15 also decreased upon SPOPL depletion when a siRNA resistant construct of SPOPL was overexpressed (*Figure 3—figure supplement 1B*). Quantitative RT-PCR revealed that EPS15 accumulation observed in SPOPL-depleted cells was not associated with increased EPS15 mRNA (*Figure 3—figure supplement 1C*). Moreover, GFP-tagged EPS15 expressed from the doxycycline (Dox)-inducible promoter was elevated at least five fold in SPOPL-depleted cells compared to RNAi-controls (*Figure 3—figure supplement 1D*). Together, these data suggested that EPS15 is degraded in a SPOPL- and CUL3-dependent manner.

Ubiquitination can specifically target a protein for degradation either through the 26S proteasome, or through lysosomal pathways (*Doherty and McMahon, 2009*; *Schreiber and Peter, 2014*). To understand which process is responsible for EPS15 degradation, we treated the GFP-tagged EPS15 cell line with either MG132 to block proteasome activity or chloroquine to stop endosome maturation (*Figure 3C*). GFP-EPS15 was stabilized upon MG132 addition in a concentration-dependent manner, but not by chloroquine. Similarly, endogenous EPS15 levels slightly increased with increasing MG132 concentrations (*Figure 3—figure supplement 1E*), indicating that EPS15 is degraded by the proteasome.

We next examined whether EPS15 is a substrate of the CRL3$^{SPOPL}$ complex. Immunofluorescence microscopy showed SPOPL-GFP in vesicle-like structures that partially overlapped with EPS15 and EGFR (*Figure 3D*). Moreover, SPOPL-GFP partially co-localized with the early endosomes marker EEA1, while it was not detected in LAMP1-containing structures. This indicated that SPOPL does not associate with lysosomes. Consistent with this localization data, endogenous immunoprecipitation revealed that EPS15 precipitates with SPOPL but not with SPOP in cell extracts (*Figure 3E*). Furthermore, recombinant GST-tagged EPS15 was purified from *E. coli* and tested for its ability to bind to purified SPOPL. SPOPL was eluted specifically with GST-EPS15 in vitro (*Figure 3F*), demonstrating direct association of the two proteins.

With evidence for in vitro and in vivo association between CRL3$^{SPOPL}$ and EPS15, we tested whether EPS15 could be ubiquitinated by the CRL3$^{SPOPL}$ complex in vitro. We incubated recombinantly purified EPS15 with ubiquitin and neddylated CUL3/RBX1 complexes, with or without the addition of SPOPL. As shown in *Figure 3G*, EPS15 was readily ubiquitinated in a SPOPL-dependent manner. Depending on the E2 enzyme - UBE2R1 (CDC34) or UBE2D1 (UBCH5) - EPS15 was poly-, mono- and di-ubiquitinated, respectively (*Figure 3G* and *Figure 3—figure supplement 1F*).

These results demonstrated that EPS15 is ubiquitinated in a CRL3$^{SPOPL}$-dependent manner in vitro and that ubiquitination leads to its proteasomal degradation.

## SPOPL binds EPS15 via a conserved motif and ubiquitinates lysine 793

EPS15 contains two ubiquitin-interacting motifs (UIM) in its C-terminal domain that serve as a hub for regulation via ubiquitination in vivo. Moreover, the EPS15 amino acid sequence predicts SPOP binding motifs (Φ-π-S-S/T-S/T, Φ = nonpolar, π = polar) (*Figure 4A* and *Figure 4—figure supplement 1A*, *Zhuang et al., 2009*).

To determine whether EPS15 is also ubiquitinated in a SPOPL-dependent manner in vivo, we prepared cell extracts from SPOPL-depleted and RNAi control cells, and used a monoclonal antibody to enrich for isopeptides containing the K-ε-GG remnant motif after trypsin digestion of ubiquitinated substrate proteins (*Kim et al., 2011*). Modified peptides were then eluted and quantified with liquid chromatography coupled to tandem mass-spectrometry (LC-MS/MS) (*Figure 4B*, *Figure 4—figure*

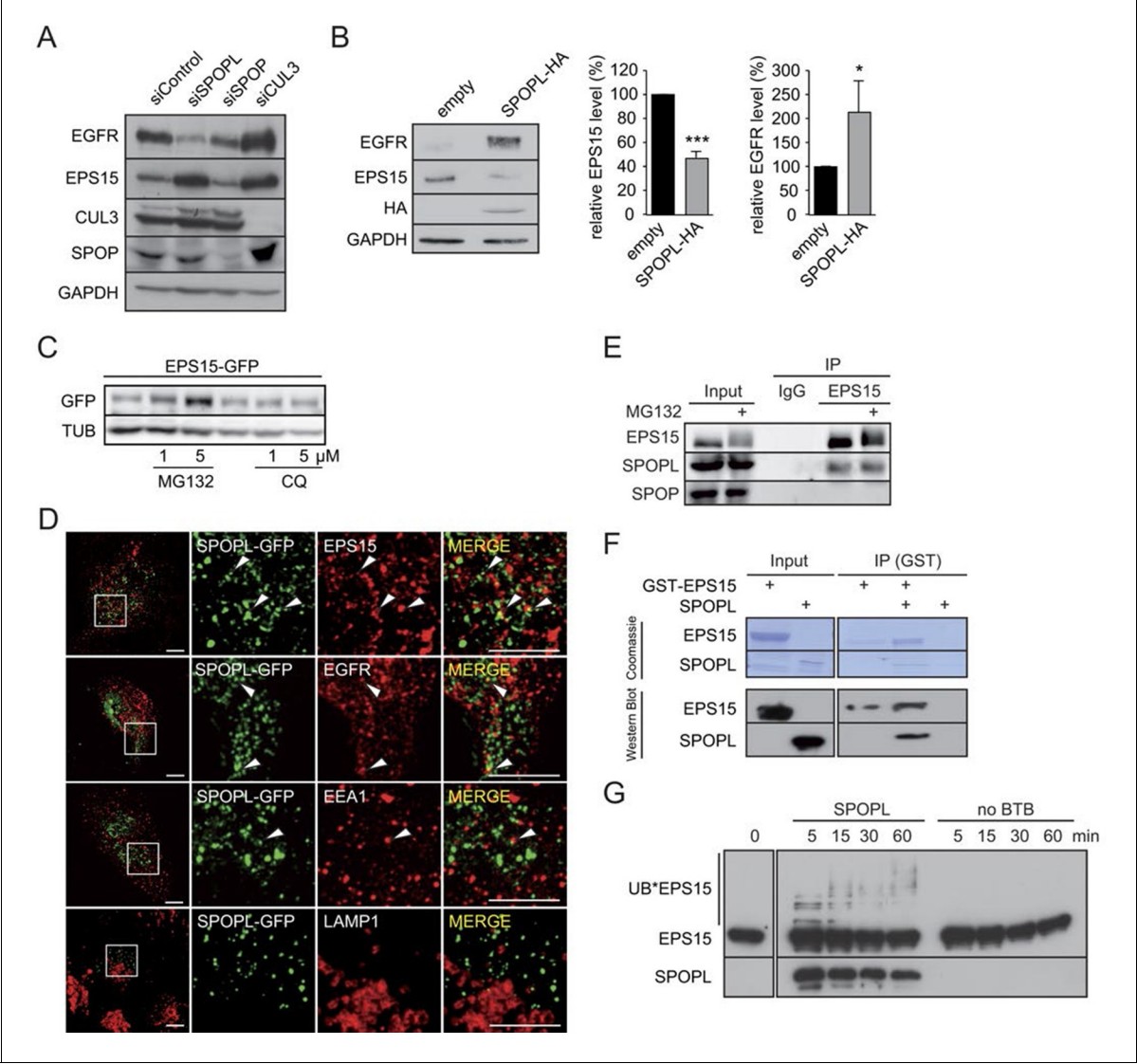

**Figure 3.** CRL3[SPOPL] targets EPS15 for proteasome-dependent degradation. (**A**) Total cell extracts prepared from HeLa cells treated with control siRNA (siControl) or RNAi oligos targeting SPOPL, SPOP or CUL3 as indicated were analyzed by immunoblotting for EGFR, EPS15, SPOP and CUL3 protein levels. GAPDH controls for equal loading. (**B**) Total cell extracts prepared from HeLa cells harboring an empty control plasmid (empty) or a plasmid overexpressing HA-tagged SPOPL were analyzed by immunoblotting for EGFR, EPS15 and SPOPL-HA protein levels. GAPDH controls for equal loading. EPS15 and EGFR levels were quantified by Image J. Data are mean + SD, N = 3. ***$p \leq 0.001$; *$p \leq 0.05$. (**C**) The levels of EPS15-GFP expressed from the doxycycline-inducible promoter were analyzed by immunoblotting of extracts prepared from HeLa cells for 40 hr with either MG132 or chloroquine (CQ). Tubulin (TUB) controls for equal loading. (**D**) A549 cells transiently transfected with a plasmid expressing SPOPL-GFP were treated with 10 µM MLN-4924 to stabilize SPOPL-GFP levels. After 6 hr, cells were fixed, stained with specific antibodies and analyzed by confocal immunofluorescence microscopy. Displayed are maximal projections of Z-stack acquisitions, fully covering cell height. Scale bar = 10 µm. Regions of interest (squares) are shown at 4x higher magnification. (**E**) Endogenous EPS15 was immunoprecipitated (IP) from HEK-293 cells using a specific antibody or unspecific IgG as control, after pretreated with 1 µM MG132 for 30 hr. EPS15 and co-precipitated proteins were eluted and analyzed by immunoblotting using specific antibodies. 40 µg of protein were loaded as input samples. (**F**) In vitro binding of recombinantly purified SPOPL to GST-EPS15 in GST pull-down experiments was analyzed by Coomassie staining (upper panel) and immunoblotting (lower panel), respectively. (**G**) In vitro ubiquitination assays. *E.coli* purified EPS15 and reconstituted CUL3-NEDD8-RBX1 were incubated at 37°C using UBE2R1 (CDC34) as the E2-enzyme and in the presence of SPOPL or without BTB adaptor (no BTB). Aliquots were taken at the indicated time points (minutes) and the presence of EPS15 and SPOPL was analyzed by immunoblotting. UB*EPS15 marks the appearance of ubiquitinated EPS15.

The following figure supplement is available for figure 3:

**Figure supplement 1.** CRL3[SPOPL] targets EPS15 for proteasome-dependent degradation.

*supplement 1B*). This analysis identified several ubiquitination sites in EPS15 including K693 and K801 that were ubiquitinated irrespective of the presence or absence of SPOPL. In contrast, ubiquitination of K793, located in the C-terminal domain of EPS15 close to the ubiquitin-interacting motifs (UIMs) (*Figure 4A*), was significantly reduced in cells lacking SPOPL.

The MATH domains of SPOP and SPOPL are very similar and, in the case of SPOP, known to be responsible for substrate recognition (*Errington et al., 2012*). To test whether SPOPL recognizes the same motif in EPS15, we expressed and purified EPS15 mutant proteins with the three serines in potential SPOPL binding pocket mutated to alanine residues (S605-607A and S744-746A). While binding of SPOPL to the EPS15$^{S605-607A}$ mutant was comparable to wild-type controls, the ability of EPS15$^{S744-746A}$ to interact with SPOPL was severely reduced (*Figure 4C*). This showed that SPOPL preferentially binds EPS15 through the conserved TSSSV motif.

To confirm that the *bona fide* SPOPL-binding motif and the SPOPL targeted lysine are relevant for turnover of EPS15 in vivo, we compared the levels of C-terminally GFP-tagged wild-type EPS15, EPS15$^{S744-746A}$ and EPS15$^{K793R}$ stably expressed in HeLa cells from a doxycycline-inducible promoter. Indeed, the steady-state levels of EPS15$^{S744-746A}$-GFP were increased at least six fold, concomitant with decreased EGFR levels, and no further increase of EPS15$^{S744-746A}$-GFP levels was detected by simultaneously depleting SPOPL (*Figure 4D*). Furthermore, we analysed the EPS15$^{K793R}$ mutant, in which in addition to the lysine 793 the neighboring lysine 788 was mutated to an arginine to prevent spurious ubiquitination. Although it showed reduced expression, when compared to wild type, no stabilization was detected after SPOPL depletion, suggesting that the lysine 793 is indeed relevant for EPS15 turnover via CRL3$^{SPOPL}$ in cells.

Furthermore, when HA-tagged SPOPL was overexpressed, degradation of the binding mutant EPS15$^{S744-746A}$-GFP and the SPOPL-ubiquitination-deficient mutant EPS15$^{K793R}$-GFP were not induced in contrast to wild-type EPS15 (*Figure 4E*). Like EPS15-GFP, EPS15$^{S744-746A}$-GFP and EPS15$^{K793R}$-GFP localized at the cell surface and to punctate intracellular structures (*Figure 4—figure supplement 1C*), implying that its interaction with SPOPL does not interfere with its subcellular localization. We concluded that CRL3$^{SPOPL}$ binds to a conserved SPOP/SPOPL binding motif in EPS15. It ubiquitinates EPS15 on lysine K793, which evidently results in proteasomal degradation of EPS15.

## SPOPL-mediated ubiquitination of EPS15 affects EGFR sorting and degradation

Next, we asked why loss of SPOPL-mediated EPS15 ubiquitination reduced the level of EGFR. We found that EGFR was more rapidly degraded after addition of EGF in SPOPL-depleted cells compared to RNAi-controls, while its internalization judging by EGFR ubiquitination and EGF uptake was similar (*Figure 5A* and *Figure 5—figure supplement 1A*). The degradation was blocked by chloroquine, which inhibits the acidification of endosomes and LYs (*Figure 5B*), confirming that EGFR was degraded in lysosomes.

To examine whether SPOPL-dependent ubiquitination of EPS15 protects EGFR from lysosomal degradation, we stably overexpressed C-terminal GFP-tagged EPS15 from the Dox-inducible promoter in RNAi-control and SPOPL-depleted cells. Overexpression of EPS15-GFP only decreased the EGFR levels in the absence of SPOPL (*Figure 5C*), implying that EPS15 accumulation alone may not be sufficient to promote EGFR degradation. Rather SPOPL-dependent ubiquitination directly regulates EPS15 activity.

To test whether the presence of EPS15 is required for lysosomal EGFR degradation in SPOPL-depleted cells, we measured EGFR levels in SPOPL-depleted cells that are simultaneously depleted for EPS15. Indeed, EGFR levels increased in cells lacking both SPOPL and EPS15, demonstrating that EPS15 is necessary to promote EGFR degradation in the absence of SPOPL (*Figure 5D*).

Finally, we investigated why loss of SPOPL-mediated ubiquitination of EPS15 increased EGFR degradation in lysosomes. EPS15 not only promotes EGFR uptake via clathrin-mediated endocytosis, but it is also localized in places of intracellular sorting and degradation (*Gucwa and Brown, 2014*; *van Bergen en Henegouwen, 2009*). We used super resolution microscopy in structured illumination mode to localize EPS15 in SPOPL-depleted cells and controls. While EPS15 at the plasma membrane was unaffected by SPOPL depletion (*Figure 5—figure supplement 1B*), EPS15 accumulated in intracellular structures upon SPOPL-depletion and increasingly co-localized with the ESCRT-0 subunit HRS (*Figure 5E*). The number and size of the EPS15-containing vesicles was unaffected

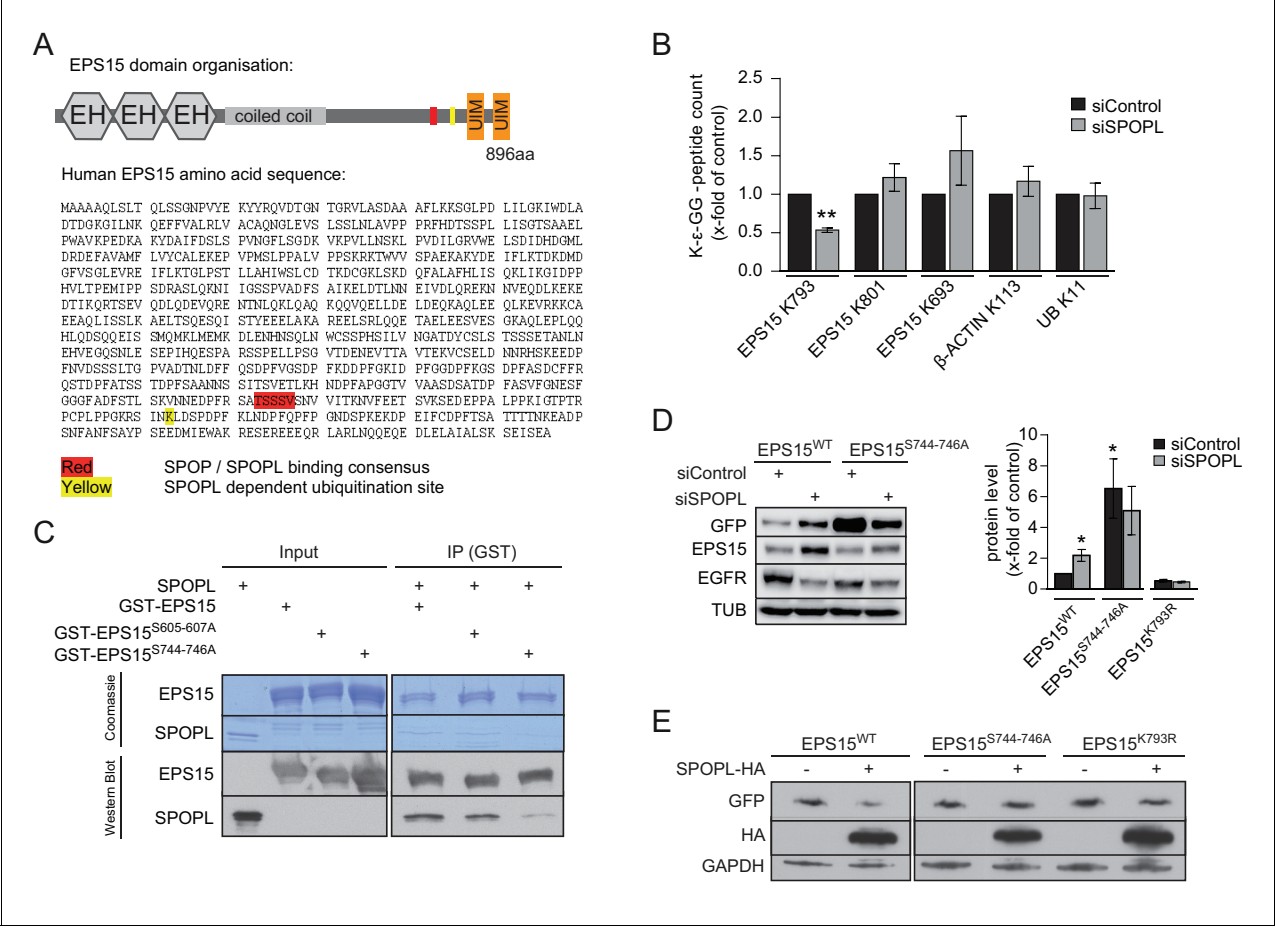

**Figure 4.** EPS15 is targeted via a SPOP/SPOPL binding consensus motif. (**A**) Cartoon of human EPS15 domain-organization and the amino-acid sequence. Indicated by color code are the SPOP/SPOPL binding site (red) and the lysine residue (yellow), which is ubiquitinated in a CRL3$^{SPOPL}$–dependent manner in vivo. In addition, the amino-terminal Ca$^{2+}$-binding EF-hand motifs (EH), the coiled-coil domain involved in dimerization and the two carboxy-terminal ubiquitin-interacting motifs (UIMs) involved in ubiquitin-binding are indicated. (**B**) EPS15 ubiquitin-profiling. Peptides containing EPS15 modification sites were quantified with LC-MS/MS after enrichment of the K-ε-GG motif from whole cell digests of HeLa cells treated with siSPOPL or siControl. Normalized precursor mass intensity profiles for EPS15 sites corresponding to K793, K801 and K693 are shown (raw data in *Figure 4—figure supplement 1B*). Quantification of the β-Actin K113 and the polyubiquitin K11 linkage peptide control for comparable enrichment. Data are mean ± SD, N = 3. **p≤0.01. (**C**) Purified SPOPL was incubated as indicated with GST-tagged wild-type EPS15 or GST-EPS15 mutants, where the predicted SPOPL binding motifs have been mutated individually (GST-EPS15$^{S605-S607A}$ and EPS15$^{S744-S746A}$, respectively), pulled down with glutathione sepharose (IP [GST]) and bound proteins were analyzed by Coomassie blue staining (upper panel) and immunoblotting (lower panels). Note that SPOPL readily binds to GST-EPS15 and GST-EPS15$^{S605-S607A}$, but this interaction is strongly reduced with the GST-EPS15$^{S744-S746A}$ mutant. (**D**) HeLa cells stably expressing GFP-tagged wild-type EPS15, the EPS15$^{S744-S746A}$ or the EPS15$^{K793R}$ mutants from a doxycycline-inducible promoter were transfected as indicated (+) with control siRNA or siRNA depleting SPOPL. The levels of EPS15-GFP, EGFR and for control tubulin (TUB) were analyzed by immunoblotting with specific antibodies. Experiments were quantified in Fiji and the EPS15 levels plotted as fold-increase compared to controls. Data are mean ± SEM, N = 4. *p≤0.05. Note that SPOPL depletion does not further increase the levels of both EPS15 mutants. (**E**) Total cell extracts were prepared from HeLa cells expressing either GFP-tagged wild-type, the EPS15$^{S744-S746A}$ mutant or the EPS15$^{K793R}$ mutant in the presence (+) or absence (-) of HA-tagged SPOPL overexpression. The levels of EPS15-GFP, SPOPL-HA and control GAPDH were analyzed by immunoblotting. Note that overexpression of SPOPL-HA is able to induce degradation of wild-type but not the EPS15$^{S744-S746A}$-GFP or the EPS15$^{K793R}$-GFP mutant.

The following figure supplement is available for figure 4:

**Figure supplement 1.** EPS15 is targeted via a SPOP/SPOPL binding consensus motif.

suggesting that SPOPL-dependent ubiquitination of EPS15 is required to remove it from ESCRT-0 complexes. Because HRS is necessary for cargo sorting into the ESCRT pathway for MVB formation (*Bache, 2003*), it is plausible that the prolonged co-localization of EPS15 with HRS in the absence of SPOPL results in enhanced EGFR sorting to the lysosome. Indeed, the interaction of EPS15 and

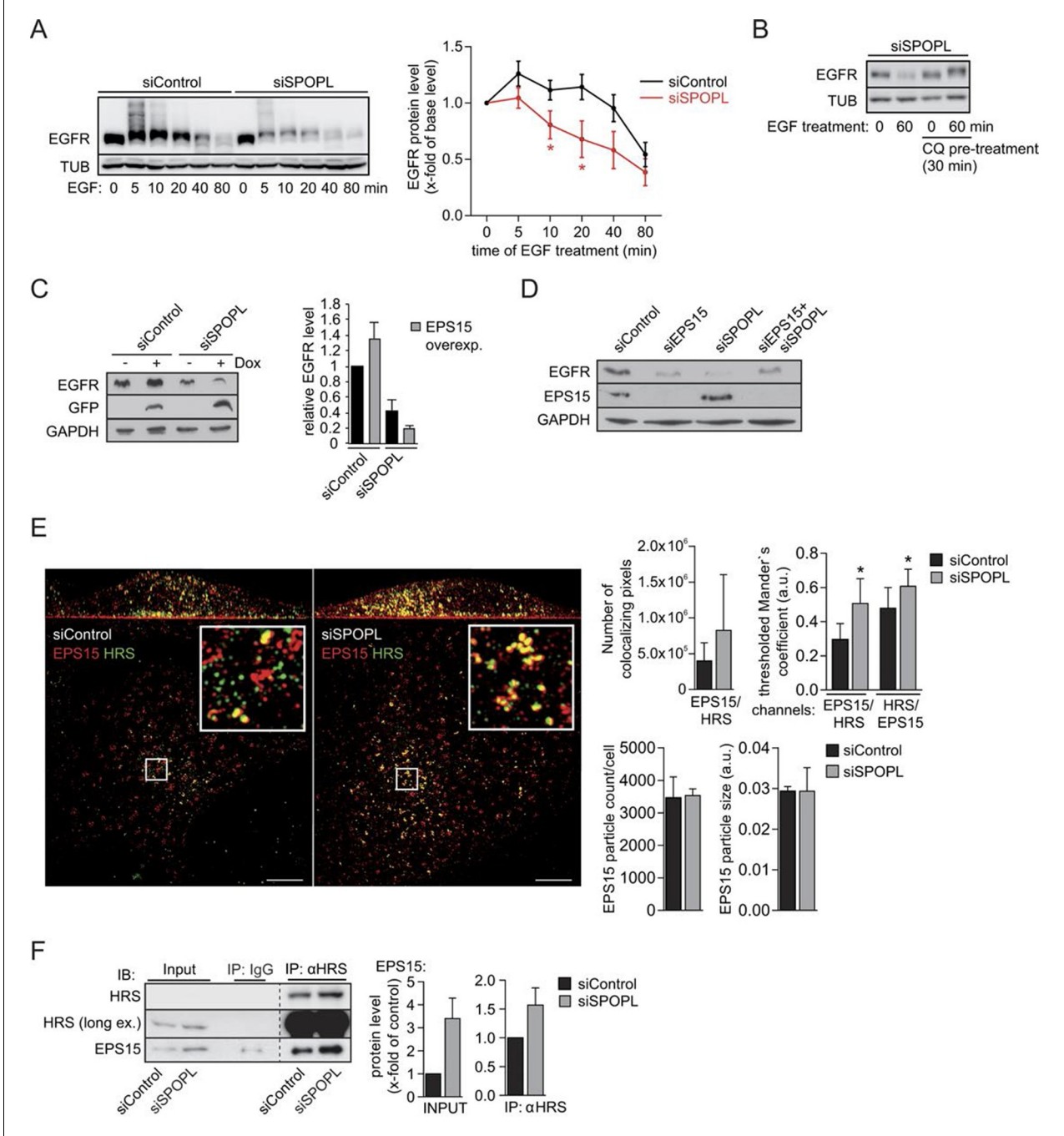

**Figure 5.** Ubiquitination of EPS15 by SPOPL regulates EGFR sorting and degradation. (**A**) HeLa cells transfected with control siRNA (siControl) or siRNA depleting SPOPL (siSPOPL) were serum-starved for 20 hr and treated with EGF (200 ng/µl) for the indicated times (minutes). EGFR levels and ubiquitination were analyzed in total cell extracts by immunoblotting. TUB controls for equal loading. EGFR levels were plotted as fold-increase compared to basal levels against the time of EGF treatment (right panel). Data are mean ± SEM, N = 5. *p≤0.05. (**B**) SPOPL-depleted HeLa cells were serum-starved for 20 hr, and pre-treated or not for 30 min with 20 µM chloroquine (CQ). EGF (200 ng/µl) was then added, and EGFR levels analyzed by immunoblotting of total cell extracts prepared at time 0 or after 60 min. TUB controls for equal loading. (**C**) Total cell extracts were prepared from HeLa cells induced (+) or not (-) to express GFP-tagged EPS15 from the doxycycline (Dox)-inducible promoter and treated with control or SPOPL siRNAs. The levels of EGFR, EPS15-GFP and for control GAPDH were analyzed by immunoblotting. EGFR levels were quantified in Image J. Data are mean + SD, N = 3. (**D**) Total cell extracts prepared from HeLa cells treated for 3 days with control siRNA or RNAi oligos targeting EPS15 and SPOPL, individually and together, were analyzed by immunoblotting for EGFR and EPS15 protein levels. Equal loading was controlled by immunoblotting for GAPDH. (**E**) Cells transfected with control siRNA (siControl) or siRNA depleting SPOPL (siSPOPL) were analyzed by indirect immunofluorescence for EPS15 (red) and HRS (green) using super resolution microscopy (SRM) in structured illumination mode. Maximal projection is shown (left panel). Scale bar = 5 µM. The

*Figure 5 continued on next page*

*Figure 5 continued*

squares are shown at 5x higher magnification in the insets. Co-localization of EPS15 and HRS as well as the number and size of EPS15-positive endosomes was quantified using Fiji (right graphs). Data are mean ± SEM, 20 > n < 10, N = 4; *p≤0.05; **p≤0.01. (F) Cell extracts were prepared from HeLa cells 72 hr after transfection with control siRNA or siRNA targeting SPOPL (siSPOPL), and incubated with control IgG or antibodies against HRS. Co-precipitated proteins (IP) were eluted and analyzed by immunoblotting (IB) for the presence of HRS and EPS15. 40 µg of protein extract was loaded as input samples (left side). The input and IP protein levels were quantified using ImageJ (right side). Data are mean ± SEM, N = 6.

The following figure supplement is available for figure 5:

**Figure supplement 1.** Ubiquitination of EPS15 by SPOPL regulates EGFR sorting and degradation.

EGFR is reduced in SPOPL-depleted cells (*Figure 5—figure supplement 1C*), while the interaction of EPS15 and HRS is stabilized (*Figure 5F*), consistent with the idea that EPS15 ubiquitination promotes EGFR stabilization. Additionally, we did not detect enhanced EGFR colocalization with recycling endosomes marked by RAB11, although RAB11 staining is elevated in siSPOPL-treated cells (*Figure 5—figure supplement 1D*). Taken together, these data suggest that loss of SPOPL-mediated ubiquitination of EPS15 favors EGFR sorting and trafficking into the lysosomal pathway via HRS (*Figure 6*).

## Discussion

In this study, we identified the BTB-adaptor SPOPL as a novel host cell factor in endosome maturation and IAV entry. After SPOPL depletion, the endocytic internalization of the exogenous cargo IAV and the cellular cargo EGF was not inhibited, the morphology and function of EEs appeared to be normal. However, the formation of LEs was compromised. RAB7 positive endosomes were enlarged and failed to acquire ILVs. We identified the endocytic adaptor protein EPS15 as a direct ubiquitination target of CRL3$^{SPOPL}$. Upon SPOPL depletion, EPS15 was enriched in endosomes co-localizing with the ESCRT-0 complex subunit HRS. Taken together, the results indicated that SPOPL together

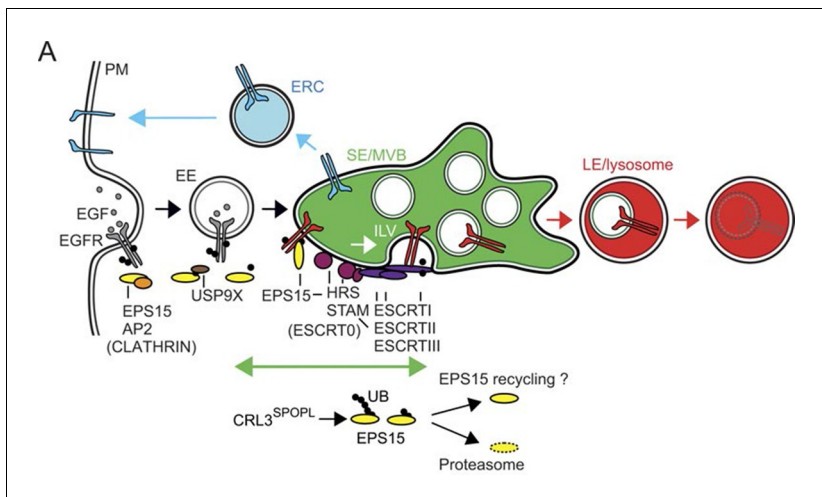

**Figure 6.** CRL3$^{SPOPL}$ ubiquitinates EPS15 at endosomes EPS15 and thereby regulates EGFR sorting and lysosomal degradation. Schematic model depicting the major EGFR trafficking routes and highlighting possible roles of EPS15 ubiquitination by CRL3$^{SPOPL}$ at endosomes. EGFR is internalized at the plasma membrane by clathrin-mediated and clathrin-independent endocytosis, which involves recognition of the ubiquitinated receptor by EPS15. EGFR is then either recycled back to the plasma membrane via the endocytic recycling compartment (ERC), or targeted for degradation into late endosomes (LE)/lysosomes after its uptake into sorting endosomes (SE)/ multivesicular bodies (MVBs) by the ESCRT machinery. Loss of CRL3$^{SPOPL}$ activity results in enhanced EGFR sorting into the degradative pathway, suggesting that CRL3$^{SPOPL}$-mediated ubiquitination of EPS15 at endosomes delays EGFR trafficking to lysosomes. For further explanation, see text.

with CUL3 regulates the endosome pathway by ubiquitinating EPS15 and inducing its degradation (*Figure 6*). The degradation of EPS15 is required for ILV formation and possibly other aspects of endosome maturation. The immature LEs can support IAV acidification and penetration by membrane fusion but not uncoating of the capsids, a process that occurs on the cytosolic surface of LEs.

## CRL3^SPOPL regulates EPS15 function at LEs via ubiquitination

EPS15 was originally identified as a substrate for the EGFR kinase that interacts with the clathrin assembly adaptors AP2 and EPSIN1 (*Benmerah et al., 1995*; *Chen et al., 1998*; *Fazioli et al., 1993*). EPS15 possesses four domains: a N-terminal domain with three EH motifs, a central coiled-coil domain involved in dimerization, and a C-terminal regulatory domain with the AP2-binding site followed by two UIM motifs (*Figure 4A*) (*van Bergen en Henegouwen, 2009*). Perturbation of EPS15 and EPSIN1 function blocks endocytosis of EGF and transferrin, demonstrating that they are part of the primary endocytic machinery at the plasma membrane (*Carbone et al., 1997*). In addition, EPS15 promotes vesicular trafficking intracellularly (*Yuan et al., 2014*). Overexpression of shorter isoforms (EPS15S, EPS15B) determines cargo recycling or degradation at endosomes (*Chi et al., 2011*; *Roxrud et al., 2008*).

EPS15 is moreover known to associate with the E3-ligases NEDD4 and Parkin (PARK2) and is mono-ubiquitinated by them (*Fallon et al., 2006*; *Polo et al., 2002*; *Woelk et al., 2006*). The putative E3 binding sites for these ligases are located in the extreme C-terminal end of the EPS15 regulatory domain close to the UIMs. The binding sites are distinct from the TSSSV motif required for interaction with SPOPL. Several ubiquitination sites have been reported in EPS15 (*Kim et al., 2011*; *Savio et al., 2016*; *Wagner et al., 2011*) and most of them are not affected by downregulation of SPOPL in vivo as observed in our mass-spectrometry analysis (*Figure 4—figure supplement 1B*). That different E3 ligases appear to ubiquitinate EPS15 in distinct sites, may have to do with its fate in different subcellular locations and/or with response to distinct signals.

Our findings support previous reports that ubiquitination inhibits the functions of EPS15. For example, a single ubiquitin in the C-terminal domain reduces binding of EPS15 to free ubiquitin thus impairing association with ubiquitinated client proteins and co-localization with EGFR on endosomes (*Hoeller et al., 2006*). Our own data shows that ubiquitination by the CRL3^SPOPL complex inhibits EPS15 function in endosome maturation (*Figure 6*). In this case, ubiquitination may control the timing and duration of EPS15 interaction with ESCRT-0, which in turn may influence ILV formation and the efficiency of lysosomal sorting and degradation of EGFR and other targets. It may also prevent the remodeling of the endosomal limiting membrane during endosome maturation, leading to loss of components needed for uncoating of incoming IAV capsids.

## EPS15 regulates EGFR trafficking at several steps

EPS15 forms a trimeric complex with the ESCRT components HRS and STAM and interacts with EGFR and HRS 30 min after EGF uptake (*Bache, 2003*; *Sigismund et al., 2005*) Upon HRS depletion, EPS15 accumulates on endosomes (*Gucwa and Brown, 2014*). Our results showed that SPOPL depletion promotes endosomal co-localization of EPS15 with HRS. This implies that EPS15 association with HRS is important for EPS15 turnover and for targeting of the EGFR to lysosomes for degradation.

That EPS15 accumulates on ubiquitin-enriched endosomes suggests that in addition to HRS it is recruited by ubiquitinated clients (*Gucwa and Brown, 2014*). A change in the phosphorylation state of EPS15 - either due to the mutation EPS15-Y850F or overexpression of the phosphatase PTPN3 – results in faster degradation of EGFR (*Li et al., 2014*). Taken together, these findings indicate that EPS15 promotes sorting of ubiquitinated EGFR at endosomes by interacting with HRS. Because upon SPOPL depletion EPS15 accumulates on HRS-containing vesicles and EGFR degradation is accelerated, we propose that the EPS15-HRS complexes actively promote sorting of EPS15 clients such as the EGFR and MET to lysosomes.

Interestingly, recent findings further demonstrate that EPS15 de-ubiquitination by USP9X affects EGFR internalization and its trafficking to lysosomes (*Savio et al., 2016*). In contrast to SPOPL depletion, the depletion of USP9X leads to slower degradation of EGFR. Therefore, it is evident that a cycle of ubiquitination / de-ubiquitination of EPS15 not only regulates the internalization pathway, but is also involved in the sorting of cargo in LEs (*Figure 6*). Ubiquitination may increase EPS15

turnover in endosomes. This may first involve EPS15 binding to its own UIM domains followed by poly-ubiquitination to target EPS15 for proteasomal degradation. While SPOPL-depletion promotes the HRS- and ESCRT-dependent fast-forward sorting route for EPS15 client cargo, other sorting processes may be inhibited due to depletion of the ESCRT processing machinery. Indeed, ILV formation is a dynamic and tightly regulated process that involves recycling of the ESCRT components by VPS4 (*Babst et al., 1998*; *Sachse, 2004*). It is evident that ILV formation depends on SPOPL, and that loss of available ESCRT complexes due to loss of CRL3$^{SPOPL}$ activity prevents the reduction in the surface area of the endosomal limiting membrane. This is reflected in both the lack of ILVs and the apparent size-increase in LEs.

## The defect in IAV entry is caused by incomplete endosome maturation

The inhibition of IAV entry after SPOPL-depletion was similar to our previous observations with CUL3-depletion (*Huotari et al., 2012*). This late penetrating virus was endocytosed normally and reached a late endosomal compartment acidic enough to induce HA-conversion and membrane fusion/hemifusion. However, the uncoating process that involves dissociation of the matrix protein M1 shell and release of the vRNPs from the endosome did not take place. Recent studies have indicated that uncoating of IAV capsids is a complex process that requires priming of the viral capsid prior to fusion by exposure to low pH and potassium ions in the lumen of endosomes as well as the function of the ubiquitin-vacuolar protein sorting system and interaction with a number of host cell factors in the cytosol (*Banerjee et al., 2014*; *Khor et al., 2003*; *Martin and Helenius, 1991*; *Stauffer et al., 2014*). While we found that acidification of IAV occurred in the absence of SPOPL and CUL3, it is possible that the exposure to an elevated potassium concentration in late endosomes did not take place. It is also possible that cellular factors associated with LE membrane and needed for uncoating, such as the E3 ligase ITCH, did not associate with immature endosomes (*Su et al., 2013*). Moreover, the EGFR level could be crucial for IAV entry, since EGFR signaling is required for efficient IAV infection. Available evidence suggests that virus binding leads to lipid-raft clustering, which activates EGFR and other RTKs and facilitates IAV uptake (*Eierhoff et al., 2010*).

## Additional CUL3 E3 ligase complexes are most likely active in the endocytic system

Given the strong homology between SPOPL and SPOP particularly in the MATH-domain, it is surprising that their subcellular localizations diverge, with SPOPL localizing to endosomes and preferentially ubiquitinating EPS15 in vivo. SPOP and SPOPL mainly differ by a 18 amino-acid insertion in SPOPL (*Errington et al., 2012*) that may be responsible for their distinct subcellular localization, which likely underlies the observed substrate specificity. SPOP and SPOPL have also been shown to form heterodimers, and in vitro experiments have suggested that SPOPL may inhibit SPOP activity by attenuating self assembly in a dose-dependent manner (*Errington et al., 2012*). It will be important therefore to identify additional SPOP and SPOPL substrates, and further test the functional interaction of SPOP and SPOPL in physiological settings.

While we could demonstrate that CRL3$^{SPOPL}$ directly ubiquitinates EPS15, it is likely that SPOPL regulates additional substrates at endosomes. That EPS15, STAM, and to a lesser extent HRS levels were increased in SPOPL-depleted cells (*Figure 2E*) raises the possibility that CRL3$^{SPOPL}$ directly or indirectly affects these ESCRT components.

Finally, it is clear that CUL3 must modify additional targets relevant for endocytic trafficking. Indeed, downregulation of SPOPL decreases EGFR levels, whereas CUL3-depletion results in greatly elevated levels of EGFR, most likely due to a reduced ability of endosomes to fuse with lysosomes. Our RNAi screen identified BTB-adaptors other then SPOPL, and a subset of them showed increased EGFR levels mimicking those defects associated with CUL3 (MG, AU and MP, unpublished results). Detailed analysis of these adaptors may shed light on additional functions of CUL3 at late stages of endocytosis.

## Materials and methods

### Cell culture and RNAi-screening

HeLa Kyoto were kindly provided by Daniel Gerlich, A549 were obtained from American Type Culture Collection (ATCC), HeLa FRT cells were a kind gift of Stephen Taylor and HEK-293 FRT were bought from Life Technologies. HeLa Kyoto, HeLa FRT and HEK-293 FRT were cultured at 37°C and 5% $CO_2$ in Dulbecco's modified Eagle Medium (DMEM, Gibco) supplemented with 10% fetal calf serum (FCS). A549 were maintained at 37°C and 5% $CO_2$ in DMEM + Glutamax (Gibco) + 10% FCS. All cell lines were not passaged longer than 3 months and on a routine basis tested negative for mycoplasma contamination.

Site-directed integration into HeLa FRT or HEK-293 FRT was achieved with the plasmid pcDNA5-FRT/TO using the Life Technologies Flp-In System. Stable cell lines were generated by transfecting cells in a 6-well plate setting with 1.8 µg pOG44 bearing the Flp recombinase cDNA and 0.2 µg of the pcDNA5-FRT/TO by Lipofectamine 2000 or 3000 transfection reagents in Opti-MEM according to the manufacturer's instructions. Cells were plated the next day into a 15 cm dish, and individual cell clones were picked and expanded after selection with 200 µg/ml Hygromycin B. Stable cell lines were cultured in 4 µg/ml Blasticidin to maintain the Tet-Repressor, and expression from the doxycycline-regulated promoter was induced by addition of 1 µg/ml doxycycline. Generation of a stable cell line expressing a shRNA construct was done similar by using the pSUPERIOR vector and selection in 1 µg/ml Puromycin.

Transient transfections of plasmid DNA were performed using Lipofectamine 2000 or 3000 (Life Technologies) according to the manufacturers' instructions with Opti-MEM. For siRNA experiments, a final concentration of 20 nM siRNA was incubated with Lipofectamine RNAimax (Life Technologies) in Opti-MEM, and successful RNAi depletion of the target protein was assessed after 72 hr.

siRNA screenings were conducted in 96-well optical bottom plates (Greiner), and transfection was performed with 20 nM final siRNA concentration and 0.1 µl of Lipofectamine RNAimax per well. Between 1500 and 3000 A540 or HeLa cells were plated and reverse transfected for 72 hr, so that they reached 50–80% confluency on the day of infection. Cells were infected with IAV X31 virus with a concentration resulting in 20–40% infection, fixed with 4% PFA after 10–14 hr, permeabilized and blocked with 0.1% Saponin, 1% BSA and 10% FCS in PBS, followed by staining with an antibody against NP (HB65, ATCC, unpurified, 1:100) in order to detect newly synthesized viral protein. Infection was scored by applying an automated imaging and infection scoring procedure, as previously described (*Banerjee et al., 2013*). Briefly, cells were imaged on a MD2 screening microscope (Molecular Devices) equipped with a Photometrics CoolSNAP HQ camera using the 10x objective (0.3 NA Plan Fluor) and an automated autofocus. Infection efficiency was quantified using a MATLAB-based protocol, which determines the ratio of NP positive cells compared to the total cell number.

### Cloning, DNA and RNA manipulations

The SPOPL and SPOP cDNA were obtained from the Orfeome collection Version 5, and subcloned using PCR into the KpnI and XhoI sites of pcDNA5-GFP or into pcDNA5-HA-Strep-Strep for C-terminal epitope-tagging. The human EPS15 cDNA was purchased from Sino Biological, while the mouse EPS15 cDNA was a kind gift of Pier Paolo Di Fiore (Milano, Italy). Both were subcloned into pcDNA5-GFP via the KpnI and XhoI restriction sites for C-terminal epitope-tagging. Site directed mutagenesis of the SPOPL and EPS15 coding region was carried out using the Quickchange protocol (Stratagene) or the 'Round the Horn' site-directed mutagenesis protocol, and the resulting constructs were verified by sequencing. For the generation of a SPOPL siRNA resistant construct, the following primers were used: 5' CAG TGT CCA CAA TTC GGG ATA CCT CGG AAA CGG CTA AAA CAG TCC 3' and 5' GGA CTG TTT TAG CCG TTT CCG AGG TAT CCC GAA TTG TGG ACA CTG 3'. For generation of the EPS15 mutants the following primers were used: 5' CTG CGC TGA CAG GTC CAG TTG CAG 3' and 5' CTG CGT CTA CAT TAA ATG GAT CTT CCTC 3', 5' CCG CTG TCA GCA ACG TAG TGA TTA C 3' and 5' CCG CTG TGG CTG AAC GAA AAG GAT C 3', 5' AGA TTG GAT TCT CCT GAT CCC 3' and 5' GTT GAT GGA TCT TCT CCC 3'.

The RAB5, RAB7 and RAB9 cDNA were obtained from the Orfeome collection Version 5 and 8. GFP-RAB5, GFP-RAB7 and GFP-RAB9 constructs were generated by subcloning the cDNA into

pcDNA5-GFP N-terminal tag via the GATEWAY cloning system. Resulting constructs were sequence-verified.

To extract mRNA for quantitative real-time PCR analysis, cells were lysed with QIAshredder columns (Qiagen), and RNA isolated with the RNeasy kit (Qiagen) following the instructions by the manufacturer. mRNA was then reverse transcribed to cDNA with Superscript II RNase H-Reverse transcriptase (Life Technologies) and random primers (Microsynth) in the presence of RNAse OUT (Life Technologies). cDNA levels were quantified using SYBR Green PCR Master mix and the Light Cycler 480 SYBR I Master (Roche). Individual samples were normalized to the human housekeeping gene GAPDH.

## Live cell microscopy, indirect immunofluorescence and thin section electron microscopy

For live cell microscopy, cells were grown in an 8-well chamber slide (LabTek), protein expression induced with 1 µg/ml doxycycline for 8–12 hr and cells were then imaged in imaging medium (Gibco, no phenol-red, $CO_2$ independent) + FCS at 37°C using an epifluorescence-based Delta Vision microscope, with 20x, 40x, 60x or 100x objectives (20x 0.45NA Ph LUCPLFLN (Long-distance dry), 40x 1.3NA DIC Oil UApo, 60x 1.4NA DIC Oil PlanApo, 100x 1.4NA DIC Oil PlanApo), a LED illumination source and a Roper CoolSnap HQ camera. GFP-RAB7 vesicle diameter was quantified manually in Image J. EPS15-GFP levels were quantified in Image J by measuring total fluorescence levels – background of a maximal projection.

For indirect immunofluorescence, cells were grown on a 20 or 12 mm coverslip, washed with PBS and then fixed in 4% PFA for 5–15 min at room temperature (RT). After extensive washing with PBS, cells were permeablized with 0.5% NP-40 in PBS for 2 min, washed 3x with PBS + 0.01% Triton-X100 (PBS-TX100) and blocked for unspecific binding by incubation for 1 hr in PBS-TX100 containing 3% BSA. The first antibody was then added for 2 hr at RT in the presence of 3% BSA in PBS-TX100, followed by secondary antibody incubation with Alexa dye-conjugated anti-rabbit and / or anti-mouse IgGs (1:1000, Life Technologies). Coverslips were mounted in Mowiol + DAPI.

For super resolution microscopy, cells were grown on 18 mm coverslips (High Precision 170 +/- 5 µm, Marienfeld) in 6-well plates (10,000 cells/6-well), transfected with control siRNA and siRNA directed against SPOPL, and prepared for immunofluorescence staining as described above. After the final wash, the cells were shortly dipped into water, mounted onto cover glasses using non-hardening mounting medium VectaShield (vector laboratories, cat.nr. H-1000), and sealed with nail polish. Acquisitions were taken at the Deltavision OMX (PlanApoN 60x /1.42NA Oil PSF) in structured illumination mode with a 4 sCMOS OMX V4 (15bit range) camera. After image reconstruction and registration, maximal projections of all acquisitions were generated in Fiji and combined in a montage. Colocalization between two channels was analyzed using the Fiji plugin 'Colocalization Threshold'. To analyze endosome number and size an automatic threshold was applied to maximal projections and analyzed using the Fiji plugin 'Particle analysis'.

For thin section electron microscopy, cells were grown on a 12 mm coverslip in 24 well plates and knockdown of target genes was accomplished by siRNA treatment as described previously. Either cells were fixed in 2.5% glutaraldehyde with 0.05 M sodium cacodylate at pH 7.2, 50 mM KCl, 1.25 mM $MgCl_2$ and 1.25 mM $CaCl_2$ for 30 min, incubated for 1 hr at RT in 2% $OsO_4$ followed by overnight incubation with 0.5% uranyl acetate at 4°C and dehydration was done by stepwise washing of coverslips in EtOH (range from 50 – 100%), followed by washing with 100% Propylenoxid (PO). Cells were then embedded in Epon and heated at 60°C over night for polymerization. Or cells were fixed in 2.5% glutaraldehyde in 0.1 M Na-Cacodylate buffer (pH 7.4) with 0.5 mg/ml Ruthenium Red for 2 hr or in 2% formaldehyde, 1.5% gutaraldehyde in 0.1 Na-Cacodylate buffer (pH 7.4) over night at 4°C. After washing, the sample stayed in 2% uranyl acetate over night at 4°C and dehydration was done stepwise in acetone (range from 50 – 100%), embedding stepwise in Epon and samples were heated at 60°C during 48 hr for polymerization.

## Viral infection and entry assays

The influenza A virus strain X31 (A/Aichi/68, H3N2) was purchased from Virapur, CA USA. All virus assays were performed in infection medium, composed of DMEM with 50 mM HEPES and 0.2% BSA, pH 6.8. All infection assays were performed in A549 or HeLa cells, using the protocol described

above. For drug treatments, cells were pretreated for 1 hr with the drugs, and virus infection was conducted in the presence of the drug. Infected cells were fixed and the infection efficiency quantified by immunofluorescence staining of NP expression as a marker of newly synthesized viral proteins.

For bypass nuclear import, influenza virus (1 µl /well of 96-well plate) was pre-bound to cells on ice for 1 hr in infection medium. Cells were then washed once with cold infection medium on ice and replaced by either infection medium (pH 6.8) or low pH infection medium (pH 5.4, buffered with 100 mM citrate buffer) to induce viral fusion with the plasma membrane. After incubation for 2.5 min at 37°C, cells were returned to ice, washed 2 times with cold infection medium to remove traces of acid, and then incubated for 30 min in STOP medium (DMEM with 50 mM HEPES, pH 7.4 supplemented with 20 mM $NH_4Cl$ to prevent further endocytic uptake of virus). Cells were then fixed, and NP expression was quantified as described above.

All IAV entry assays have been carried out essentially as described (*Banerjee et al., 2013*). Imaging was performed with the MD screening microscope (Molecular Devices) using the 10x or 20x objectives, and the images quantified using MATLAB-based Cell Profiler modules (*Banerjee et al., 2013*).

For binding assays, cells were incubated with the virus inoculum (0.75 µl/well of 96-well plate), on ice for 1 hr in infection medium, washed 3x with cold PBS and fixed with 4% PFA. Bound virus particles were stained with a polyclonal antibody against IAV (PINDA, polyclonal, rabbit, 1:500), visualized with a fluorescently labeled secondary antibody.

To assess IAV endocytosis, virus inoculum (0.75 µl/well of 96-well plate) was bound to cells on ice for 1 hr in infection medium, after which inoculums were removed and the cells incubated for 30 min in infection medium at 37°C. Cells were fixed with 4% PFA, and non-internalized virus particles blocked with a polyclonal PINDA antibody (1:500) against IAV prior to permeabilization. Negative controls were directly fixed after the virus-binding step and termed fixation control. Cells were then permeabilized with 0.1% saponin and stained with an antibody against HA (H3, monoclonal, mouse 1:100) to detect internalized particles, followed by fluorescently labeled secondary antibody detection.

Acidification assays were essentially done as the endocytosis assay, except that the cells were incubated longer at 37°C. In brief, virus inoculum (0.75 µl/well of 96-well plate) was bound to cells on ice for 1 hr in infection medium, and then incubated at 37°C with warm infection medium for 1 hr. Cells were fixed with 4% PFA, and stained with A1 antibody (1:1000, monoclonal) specifically recognizing the acidified conformation of HA (*Webster et al., 1983*). Bafilomycin A treatment and siRNA against the vATPase subunit vATP6V1B2 were used as negative controls in these assays.

For uncoating assays, the virus inoculum (1.2 µl/well of 96-well plate) was bound to cells on ice for 1 hr in infection medium, after which cells were incubated for 2.5 hr at 37°C with warm infection medium containing 1 mM cycloheximide (CHX) to block viral protein synthesis. Cells were fixed with 4% PFA, and stained with antibody against M1 (HB64, ATCC, 1:250), followed by fluorescently-labeled secondary antibodies detection. Bafilomycin A treatment or siRNA against the vATPase subunit vATP6V1B2 were used as negative controls. Cytoplasmic dispersion of M1 and other features characteristic for viral uncoating were detected and quantified using the Advanced Cell Classifier program (www.cellclassifier.org).

Nuclear import assays were conducted as the uncoating assay with slight modifications. In brief, after virus binding, cells were incubated for 5 hr at 37°C with warm infection medium containing 1 mM cycloheximide (CHX), which was replaced with fresh CHX after 2.5 hr. Bafilomycin A treatment and siRNA against the vATPase subunit vATP6V1B2 were used as negative controls. Cells were fixed with 4% PFA, and incubated with antibody against NP (HB65, ATCC, unpurified, 1:10), followed by fluorescently-labeled secondary antibody detection and DAPI staining (1:5000). Moreover, cells were stained with WGA (1:200 in PBS) prior to permeablization. Nuclear import of NP was quantified using the MATLAB-based infection scoring program and cells were counted as positive, if a certain signal threshold was achieved in the nucleus.

For FACS-based IAV fusion assays, IAV X31 stocks were diluted in PBS to 0.1 mg/ml and labeled for 1 hr at RT with R18 and SP-DiOC18 at final concentrations of 0.2 mM. The labeled virus particles were filtered through a 0.22 µM-pore filter (Millipore) and stored at 4°C in the dark. After binding (50 µl labeled virus + 150 µl infection medium per well/ 24 well plate setting) at 4°C for 1 h, internalization for 1 hr and fixation with 4% PFA, cells were washed 3x in FACS buffer (20 mM EDTA, 2%

FCS in PBS) and analyzed using a FACS Calibur instrument for red and green signal. The red-to-green ratio was quantified using FlowJo 7.6. Bafilomycin-A-treated cells were used as negative controls showing internalization signal of the virus, but no fusion events, which would be detected as a green signal.

## EGF uptake

Cells were treated with siRNA for 72 hr, followed by serum starvation for at least for 4 hr or over-night by exchanging medium to starvation medium (DMEM without FCS). Then, cells were incubated with a final concentration of 100 ng/ml EGF (EGF-488, Life Technologies) in starvation medium for 1 hr at 4°C to allow binding to the EGF receptors. Internalization was achieved by incubating cells for 0–90 min at 37°C, followed by a short acid wash, fixation with 4% PFA and counterstaining with DAPI and WGA to visualize DNA and the cell shape, respectively. Cells were imaged on a MD screening microscope (Molecular Devices) with the 20x objective (0.75 NA S Fluor), and uptake quantified with a MATLAB-based Cell Profiler module measuring the number of EGF-containing particles per cell and the fluorescent intensities of single particles.

## Cell extracts, SDS-PAGE, immunoblotting and immunoprecipitation assays

Cells were washed with PBS, scraped off from the dish, and centrifuged for 5 min at 1200 rpm. Pellets were resolved in extraction buffer (20 mM Tris pH 7.5, 100 mM NaCl, 20 mM β-glycero-phosphate, 5 mM $MgCl_2$, 0.2% NP-40, 10% glycerol, 1 mM NaF, 0.5 mM DTT, complete Protease inhibitor mix tablet, 10 mM 1,10-Phenantroline), incubated 20 min on ice, and centrifuged for 10 min at 7500 rpm at 4°C in order to pellet bulk DNA and RNA, or treated with Nuclease for 20 min. The protein concentration was measured by Bradford assay and samples were equalized.

For SDS-PAGE and immunoblotting, 4x LDS sample buffer + 1 mM DTT was added, the extracts boiled and proteins separated by SDS-PAGE using 7–14% gradient polyacrylamide gels. For immunoblotting, proteins were transferred to a PVDF membrane (Millipore) using a semi-dry blotting device. Membranes were blocked for 30–60 min with 5% non-fat milk resuspended in PBS supplemented with 0.1% Tween 20 (PBS-T), followed by incubation with primary antibodies for 2 hr at RT or over night at 4°C. Membranes were then washed 3 times with PBS-T and incubated for 60 min with HRP-conjugated secondary antibodies resuspended in 5% milk-PBS-T. Membranes were washed three times with PBS-T and incubated with ECL solution (100 mM Tris pH 8.5 + Luminol and Coumaric acid) or with SuperSignal West Femto chemiluminescent substrate (Thermo Fisher Scientific).

For immunoprecipitation experiments, cell pellets were resolved in IP buffer (10 mM Tris pH 7.5, 100 mM KCl, 2 mM $MgCl_2$, 0.5% NP-40, 300 mM Sucrose, 10 mM β-glycero-phosphate, 0.2 mM $Na_3VO_4$, complete Protease Inhibitor tablet, 0.5 mM DTT, 1 mM PMSF, 1 µg/ml Leupeptin, 1 µg/ml Pepstatin, 1 mM NaF, 10 mM NEM, 10 mM 1,10-Phenantroline) and broken using a 27G syringe on ice. Cell extracts were cleared by centrifugation for 10 min at 10,000 rpm in a table top centrifuge. Supernatants were incubated with Strep-Tactin sepharose (IBA) or HA.7-coupled beads (Sigma) for 1 – 2 hr rotating at 4°C. After extensive washing with IP buffer, bound proteins were eluted with 100 mM Glycine pH 2.0, and analyzed by SDS-PAGE and immunoblotting. For immunoprecipitations of endogenous EPS15, extracts adjusted to the same volume and protein concentration were incubated with 10 µg anti-EPS15 antibody (Santa-Cruz, sc-534, rabbit) or anti-rabbit control IgGs for 2 hr, followed by the addition of 40 µl protein G sepharose for an additional hour. Resin was then washed 6 times with IP buffer, bound proteins eluted by boiling in 40 µl of two fold SDS sample buffer and analyzed by SDS-PAGE and immunoblotting.

## Endosome purification and fractionation

HeLa cells were harvested and resuspended in 6 ml hypotonic swelling buffer (20 mM sucrose, 20 mM HEPES NaOH pH 7.0, 10 mM KCl, 5 mM $MgCl_2$, 10 mM $CaCl_2$, 5 mM EGTA). After 10 min, cells were lysed with 30 strokes in a Dounce homogenizer using a tight pestle, and swelling was stopped by the addition of 6 ml two fold homogenization buffer (500 mM sucrose, 20 mM HEPES NaOH pH 7.0, 200 mM KCl, 5 mM $MgCl_2$, 10 mM $CaCl_2$, 5 mM EGTA). Lysates were centrifuged at 1000 g for 10 min to obtain the post-nuclear supernatant (PNS), which was further centrifuged at

13,000 g for 30 min. To purify endosomal structures, the resulting supernatant was centrifuged at 100,000 g for 1 hr, and the pellet carefully solubilized in 600 µl homogenization buffer (250 mM sucrose, 20 mM HEPES NaOH pH 7.0, 100 mM KCl, 5 mM MgCl$_2$, 10 mM CaCl$_2$, 5 mM EGTA). Insoluble particles were removed by short centrifugation and the supernatant loaded onto a 5–20% continuous OptiPrep gradient (PROGEN Biotechnik) with a total volume of 12 ml, poured according to the manufacturer's instructions using homogenization buffer for dilution. The gradient was centrifuged at 67,000 g for 18 hr, 12 fractions were collected with a pipette and proteins precipitated with 15% TCA for 1 hr. Fractions were centrifuged at 13,000 g for 1 hr and protein pellets dissolved in SDS-sample buffer for analysis by SDS-PAGE and immunoblotting.

To compare protein content of cytosolic and endosomal fractions HeLa cells (10 cell culture dishes Ø 15 cm) were lysed in 10 ml of isoosmotic homogenization buffer (0.25 M sucrose, 10 mM triethanolamine, 10 mM acetic acid pH 7.8, 1 mM EDTA, 10 mM β-glycero-phosphate, 0.2 mM Na$_3$VO$_4$, complete Protease Inhibitor tablet, 1 mM NaF, 10 mM 1,10-Phenantroline) and fractionated as described above. A lysate sample was removed, mixed with 4x LDS sample buffer and used as input control for western blotting. After final centrifugation at 100,000 g the supernatant (10 ml) was transferred to a new tube and the pellet fraction was resuspended in 10 ml of homogenization buffer. Samples of both fractions were removed, mixed with 4x LDS sample buffer and analyzed via western blotting.

For the analysis of SPOP and SPOPL via western blotting supernatant and pellet samples were up-concentrated. The pellet sample was resuspended in only 10% of the original volume and part of the supernatant sample was precipitated with 20% of trichloroacetic acid (TCA) for 30 min, centrifuged for 30 min at 13,000 g and resuspended in 10% of the original volume. 4x LDS sample buffer was then added to samples for western blot analysis. Ten fold more input sample was loaded for the up-concentrated samples to allow relative comparison with the other markers.

## EGFR degradation assay

Cells treated with siRNAs for 72 hr were incubated in starvation medium from the 2nd day onwards, and stimulated by adding 200 ng/ml unlabeled EGF. After the times indicated, cells were washed with icecold PBS, scraped off the dish, centrifuged for 5 min at 1200 rpm and the pellets frozen in liquid nitrogen. The pellets were then resolved in extraction buffer (20 mM Tris pH 7.5, 100 mM NaCl, 20 mM β-glycero-phosphate, 5 mM MgCl$_2$, 0.2% NP-40, 10% glycerol, 1 mM NaF, 0.5 mM DTT, complete Protease inhibitor mix tablet, 10 mM 1,10-Phenantroline), incubated 20 min on ice, and centrifuged for 10 min at 7500 rpm at 4°C to pellet bulk DNA and RNA, or treated with Nuclease for 20 min. The protein concentration was measured by Bradford and equalized, before adding 4x LDS sample buffer + 1mM DTT and boiling. EGFR degradation was analyzed by immunoblotting.

## Recombinant protein expression and purification

SPOPL was expressed from a modified pET17 plasmid, bearing a PreScission-cleavable His-StrepII[2x]-SUMO N-terminal solubility tag (kind gift of Anne Schreiber), in the *E.coli* Rossetta strain growing in autoinduction medium at 16°C over night. Cells were collected by centrifugation and the pellet resolved in lysis buffer (500 mM NaCl, 50 mM HEPES pH 8.0, 2 mM DTT, 5% glycerol, 1 mM EDTA and complete Protease Inhibitor cocktail tablet, 1 mM PMSF, 1 µg/ml Leupeptin, 1 µg/ml Pepstatin, 1 mg/ml lysozyme, nuclease, benzamidine). After sonication and ultracentrifugation, the supernatant was loaded on a Streptactin Superflow column (Qiagen), bound protein washed and eluted with washing buffer (200 mM NaCl, 50 mM HEPES pH 8.0, 2 mM DTT, 5% glycerol) containing 2.5 mM d-Desthiobiotin (Sigma). PreScission protease was added to remove the His-StrepII[2x]-SUMO-tag over night at 4°C. The eluate was cleaned over a GST- and His- column (GE Healthcare) to separate SPOPL from the cleaved tag and the GST-tagged PreScission. The eluate was concentrated to 1 ml using Amicon concentrator tube with a 30 kD cut off, and fractionated on a Superose 6 size exclusion column using an Äkta Pure system (GE Healthcare). SPOPL elution fractions with correct molecular weight were pooled, frozen in liquid nitrogen and stored at -80°C.

EPS15 was expressed from a pGEX-6P1 plasmid bearing a GST-tag in the *E.coli* Rossetta strain growing in autoinduction medium supplemented with 4 µg/ml CaCl$_2$ over night at 16°C. Cells were collected by centrifugation and pellets resolved in lysis buffer as described above. After sonication and ultracentrifugation, the supernatant was loaded on a GST-column (GE Healthcare), bound

protein washed and eluted with washing buffer (200 mM NaCl, 50 mM HEPES pH 8.0, 2 mM DTT, 5% glycerol) containing 20 mM reduced glutathione. PreScission protease was added to remove the GST-tag over night at 4°C, and cleaved GST was separated together with GST-tagged PreScission using a GST HiTrap column (GE Healthcare). The eluate was concentrated using an Amicon concentrator with a 100 kD cut off, and fractionated on a Superose 6 size exclusion column equilibrated in buffer (150 mM NaCl, 20 mM HEPES pH 7.8, 2 mM DTT, 2% glycerol). EPS15 fractions with correct molecular weight were pooled, frozen in liquid nitrogen and stored at -80°C.

## SPOPL binding assay and in vitro ubiquitination assays

For binding assays, glutathione sepharose 4B (GE Healthcare) aliquots were equilibrated in binding buffer (150 mM NaCl, 20 mM HEPES pH 7.8, 2 mM DTT, 2% glycerol) and incubated with equimolar amounts of GST-EPS15 and untagged SPOPL and incubated for 15 min rotating at RT. Beads were washed 4x at RT with binding buffer, and bound proteins eluted with either 20 mM glutathione in binding buffer or 100 mM Glycine pH 2.0 at RT. Eluates were immediately mixed with 4x LDS and DTT, boiled and analyzed by Coomassie Staining or immunoblotting.

Cullin-3 and RBX1 used for in vitro ubiquitination reactions were purified together as a complex from insect cells, and neddylated in vitro using purified components as described previously (*Enchev et al., 2012*; *Orthwein et al., 2015*). 0.5 µM CUL3-NEDD8-RBX1, 0.6 µM SPOPL, 0.2 µM UbE1 (Boston Biochem), 0.7 µM E2 (CDC34 or UBCH5B, produced as described in *Enchev et al., 2012*), 50 µM Ubiquitin (Boston Biochem) and 2 µM EPS15 were incubated in the presence of ATP for various times at 37°C. Reactions were stopped by the addition of 4x LDS with 1 mM DTT, boiled and analyzed by SDS-PAGE and immunoblotting.

## Ubiquitin-remnant K-$\varepsilon$-GG immunoaffinity profiling

HeLa cells treated for 72 hr with siControl oligo's or siRNA depleting SPOPL (20 confluent 15 cm dishes per condition) were scraped in their media and centrifuged at 4°C for 3 min at 300 rcf (g). Cell pellets were washed with ice-cold PBS and snap frozen in liquid nitrogen prior to lysis and digestion. After quickly thawing, pellets were resuspended in Urea lysis buffer (ULB) containing 9 M urea, 50 mM Ammonium Bicarbonate, 1 mM sodium orthovanadate, 2.5 mM sodium pyrophosphate, and 1 mM β-glycerophosphate such that the final protein concentration would be less than 5 mg/ml. A Branson 250 tip sonicator was used for lysis (power output of 15, duty cycle of 70%, 3 rounds sonication, cooling on ice between cycles) and the lysates were cleared by centrifugation at 20,000 rcf (g) for 15 min at 15°C. Reduction with 10 mM tris(2-carboxyethyl)phosphine (TCEP) was performed at RT for 30 min followed by alkylation with 20 mM iodoacetamide with the pH maintained at 7.5 and incubation for 30 min at RT protected from light. Samples were then diluted to 4 M urea with 100 mM ammonium bicarbonate buffer (pH 8) and digested with LysC (1:100) in a shaking incubator at 37°C for 4 hr. Finally, each digest was diluted to a final concentration of 1 M urea with 100 mM ammonium bicarbonate and sequencing grade trypsin was added at 1:100 dilution for overnight digestion at 37°C in a shaking incubator. The next morning, the trypsin reaction was stopped by formic acid (FA) addition to a final concentration of 1% (pH <3) and samples were placed on ice for 15 min for precipitation. Lysates were centrifuged at 2000 rcf (g) for 15 min at RT, desalted on c18 columns with 20 mg capacity and bound peptides were washed sequentially with 1 ml, 5 ml, and 6 ml of 0.1% FA in dH$_2$O followed by 2 ml of 2% acetonitrile (ACN) in 0.1% FA in dH$_2$O. Finally, peptides were eluted with 3 x 3 ml 50% ACN in 0.1% FA in dH$_2$O and lyophilized for 48 hr to remove all of the FA.

Immunoprecipitation of K-ε-GG modified peptides was carried out with an antibody-based system essentially as described in the Cell Signaling Technology, Inc. standard protocol for the PTMScan UbiScan kit. Eluted K-ε-GG peptides were desalted on self-packed StageTips as follows: StageTips were equilibrated with 50 µl 50% ACN in 0.1% trifluoroacetic acid (TFA) in dH$_2$O (1x) followed by 50 µl dH$_2$O with 0.1% TFA (2x). Samples were loaded in two steps loading 50 µl each time and passing the solution through with a centrifuge at 2000 rcf (g) for 30 s. The bound peptides were washed with 55 µl dH$_2$O with 0.1% TFA (2x) and eluted with 10 µl 50% acetonitrile (ACN) in 0.1% TFA in dH$_2$O followed by a second elution pooled with the first of 20 µl 50% acetonitrile (ACN) in 0.1% TFA in dH$_2$O. Solvents were evaporated in a thermal vacuum centrifuge to ~ 1 µl and the final peptide product was resuspended in 10 µl 2% ACN in 0.1% TFA in dH$_2$O for analysis by LC-MS/MS

on a Thermo Fisher Scientific Q-Exactive Plus in data-dependent analysis (DDA) mode. For this, peptides were separated on an EASY-nLC 1000 (Thermo Fisher Scientific) coupled to a 15 cm fused silica emitter (75 μm inside diameter). The analytical column was packed with ReproSil-Pur C18-AQ 120 Å and 1.9 μm resin (Dr. Maisch HPLC GmbH) and peptides were separated with a 120 min gradient that went from 2%–35% acetonitrile in water with 0.1% formic acid. A top 12 method was used for DDA acquisition within a mass range of 300–1700 m/z using higher energy collisional dissociation (HCD) for peptide fragmentation resulting in high resolution MS/MS data for analysis.

Raw files were converted to mzXML using msconvert and searched with Comet as part of the TransProteomicPipeline version 4.7.0. Peptides were filtered for a 1% false discovery rate based on their peptide prophet scores and Skyline version 2.6.0 was used to integrate the area under the MS1 chromatography peaks to generate label-free quantification information. To do this, chromatographic traces were aligned for each quantified peak and integration areas were refined manually for multiple peptides of interest. For this, chromatographic traces were aligned and peak areas selected manually for the peptides of interest.

In this study, three modified EPS15 peptides were analyzed with the following sequences: Lys 693 (K693) IDPFGGDPFK*GSDPFASDCFFR, Lys 793 (K793) RSINK*LDSPDPFK, and Lys 801 (K801) LDSPDPFK*LNDPFQPFPGNDSPK. As controls, a peptide containing modified Lys 113 (K113) of β Actin (VAPEEHPVLLTEAPLNPK*ANR) and the polyubiquitin linkage peptide corresponding to Lys 11 (K11) of ubiquitin (TLTGK*TITLEVEPSDTIENVK) were quantified (K* = K-ε-GG modification; C = carbamidomethyl-modified cysteine). The results reported here represent biological triplicate experiments of RNAi depletion of SPOPL compared to siControl performed on separate days. Each data point within each biological replicate represents the results from technical replicate LC-MS/MS injections. Normalization for MS intensity variations between runs was performed by setting the value of the siControl for each individual biological replicate pair to 1 and scaling the RNAi depletion condition to this set value. Final reported results are the mean ± SD.

## Antibodies, drugs and siRNAs

The following antibodies were used in this study: anti-Actin (Millipore, MAB1501R, 1:1000 in WB), anti-AP2 (adaptin alpha)/(BD Transduction, 610502, 1:1000 in WB), anti-CUL3 (described in (*Sumara et al., 2007*), 1:1000 in WB), anti-Clathrin (BD Biosciences, 610499, 1:3000 in WB), anti-EEA1 (BD Transduction 610457, 1:2500 in WB, 1:200 in IF), anti-EGFR (Millipore, 06–847, 1:1000 for WB, 1:500 in IF; or Biolegend, 352901, 1:200 in IF), anti-EPS15 mouse (BD Transduction, 610807, 1:250 in WB, 1:50 in IF), anti-EPS15 rabbit (Santa Cruz, sc-534, 1:200 in WB, 1:50 in IF), anti-GAPDH (Sigma G8795, 1:5000 in WB), anti-GFP (Roche, 11 814 460 001, 1:1000 in WB), anti-H4 (Abcam, ab16483, 1:1000 in WB), anti-HA.11 (Covance, MMS-101R, mouse, 1:1000 in WB and IF), anti-HRS (Sigma, WH0009146M1, 1:1000 in WB, 1:500 in IF), anti-LAMP1 (Santa Cruz, sc-20011 (mouse), 1:1000 in WB, 1:200 in IF or Abcam (rabbit) ab24170, 1:1000 in WB), anti-RAB11 (Sigma, R5903, 1:200 in IF), anti-SPOPL (rabbit polyclonal antibody raised against the N-terminal SPOPL peptide MSREPTPPLPGDMST+C and a SPOPL middle region peptide CKDGKNWNSNQATDIM and affinity-purified with the recombinantly expressed and purified SPOPL protein, 1:500 in WB), anti-SPOP (Abcam, ab81163, 1:500 in WB), anti-STAM (Santa Cruz, SC-133093, 1:500 in WB), anti-Tubulin (Sigma, T5168, 1:10000 in WB), anti-mouse IgG (H+L) CF405S (Biotum, BI-20080, 1:1000 in IF), anti-rabbit IgG (H+L) Alexa Fluor568 (Thermo Fisher Scientific, A-11036, 1:1000 in IF) and anti-K-ε-GG antibody (Cell Signaling Technology, cat. no. 5562).

Furthermore, following antibodies were used in *Figure 2E*: anti-Calnexin (kindly provided by Ari Helenius), anti-Caveolin 1 (Santa Cruz, sc-894, 1:500 in WB), anti-CHMP6 (Sigma, SAB2701297, 1:1000 in WB), anti-EPS15R1(Novus Biologicals, NB100-88149, 1:500 in WB), anti-EPSIN1 (Santa Cruz, sc-8673, 1:500 in WB), anti-HER2 (Cell Signaling, 2165, 1:500 in WB), anti-IGF1R (R&D Systems, MAB301, 1:1000 in WB), anti-LC3 (Novus Biologicals, NB600-1384, 1:1000 in WB), anti-MET (Santa Cruz, sc-10, 1:1000 in WB), anti-RAB7 (Cell Signaling, 9367, 1:1000 in WB), anti-SQSTM1 (American Research Products Inc., 03-GP62-C, 1:1000 in WB), anti-TSG101 (Novus Biologicals, NB200-112, 1:500 in WB), anti-VEGFR3 (Millipore, MAB3757, 1:500 in WB)

The following chemicals were used in this study: MG132 (Sigma, 1 – 5 μM final conc.), Bafilomycin A (BafA, Sigma, 50 nM final conc.), Cyclohexamide (CHX, Sigma, 10 μM – 1 mM final conc. dependent on assay), MLN-4924 (Active Biochem, 10 μM final conc.) and Chloroquine (CQ, Sigma, C6628, 20 μM final conc.).

All siRNAs were ordered from Qiagen, and their Allstar-negative and Allstar-death siRNAs were used for controls. The following siRNAs were used to specifically deplete the indicated proteins:

| Name | Target sequence |
| --- | --- |
| siCUL3 | AACAACTTTCTTCAAACGCTA |
| siSPOPL_1 | CAGTTTGGCATTCCACGCAAA |
| siSPOPL_2 | GGCCTTAAATTATCTTCAATT |
| siSPOPL_3 | GGTGCCTGAGTGTCGTCTATT |
| siSPOP 1 | TCAGTTTATCATTTGCTCC |
| siSPOP_2 | GGCTCACAAGGCTATCTTATT |
| siSPOP_3 | GGAGGAAAUGGGUGAAGUCAU |
| siEPS15 | AAACGGAGCTACAGATTAT |
| sivATPase | CACGGTTAATGAAGTCTGCTA |

## Statistical analysis

Data are represented as the means of at least triplicate experiments + standard deviation (SD), except for data in *Figure 4D*, *5A,E and F*, as well as *Figure 5—figure supplement 1B and C*, for which the results are shown as means + standard error of the mean (SEM). 'N' represents the number of replicates, and 'n' the number of measured cells. One-tailed Student's t-tests with unequal variance assumption were performed to compare the datasets for statistical significance when appropriate.

## Acknowledgements

We are grateful to K Boucke and R Mancini for the EM studies; T Courtheoux for help with super-resolution microscopy, the ETH ScopeM facility for imaging support, P Horvath for help with image analysis; O Oros and A Ragheb for technical support, and PP Di Fiore, P De Camili, S Taylor, D Gerlich and A Schreiber for reagents. We thank A Spang and members of the Helenius and Peter laboratories for helpful discussions. We are grateful to A Smith for critical reading of the manuscript. MG and NM-S were supported by Boehringer-Ingelheim-Fonds stipends, AU by the Horten-Stiftung, RIE by ETH pioneer and Marie Curie fellowships. Work in the Greber laboratory was supported by the Swiss National Science Foundation (SNF) (project number 310030B_160316) and SystemsX.ch (project VirX). The Peter and Helenius laboratories were supported by funding from the European Research Council (ERC), the SNF and ETH Zürich. In addition, the Peter laboratory was supported by Oncosuisse and a GRL grant from the Korean Government, and the Helenius laboratory by National Institutes of Health (NIH) (1UO1AI074523) and Marie Curie Initial Training Networks (ITN) grants.

## Additional information

### Funding

| Funder | Author |
| --- | --- |
| Schweizerischer Nationalfonds zur Förderung der Wissenschaftlichen Forschung | Michaela Gschweitl<br>Anna Ulbricht<br>Christopher A Barnes<br>Radoslav I Enchev<br>Ingrid Stoffel-Studer<br>Nathalie Meyer-Schaller<br>Jatta Huotari<br>Urs F Greber<br>Ari Helenius<br>Matthias Peter |

| European Research Council | Ari Helenius<br>Matthias Peter |
|---|---|
| Eidgenössische Technische<br>Hochschule Zürich | Radoslav I Enchev<br>Ari Helenius<br>Matthias Peter |
| SystemsX | Urs F Greber |
| Oncosuisse | Matthias Peter |
| National Institutes of Health | Ari Helenius |
| Marie Curie Initial Training<br>Network | Radoslav I Enchev |
| Boehringer Ingelheim Fonds | Michaela Gschweitl<br>Nathalie Meyer-Schaller |
| Horten Stiftung | Anna Ulbricht |

The funders had no role in study design, data collection and interpretation, or the decision to submit the work for publication.

## Author contributions

MG, Conception and design, Acquisition of data, Analysis and interpretation of data, Drafting or revising the article, Contributed unpublished essential data or reagents; AU, Acquisition of data, Analysis and interpretation of data, Drafting or revising the article, Contributed unpublished essential data or reagents; CAB, Conducted the ubiquitination profiling via mass spectrometry and its data analysis, Acquisition of data, Analysis and interpretation of data, Drafting or revising the article; RIE, Purified protein complexes and assisted in ubiquitination assay design and analysis, Analysis and interpretation of data, Drafting or revising the article, Contributed unpublished essential data or reagents; IS-S, Assisted in cloning and cell culturing, Acquisition of data, Contributed unpublished essential data or reagents; NM-S, JH, Performed the siRNA screenings, Acquisition of data, Analysis and interpretation of data, Drafting or revising the article, Contributed unpublished essential data or reagents; YY, Contributed to influenza A virus assays and conducted revisions, Acquisition of data, Analysis and interpretation of data, Contributed unpublished essential data or reagents; UFG, Enabled the EM studies with help of K. Boucke, Acquisition of data, Analysis and interpretation of data, Drafting or revising the article; AH, MP, Conception and design, Analysis and interpretation of data, Drafting or revising the article

## Author ORCIDs

Matthias Peter, http://orcid.org/0000-0002-2160-6824

# Additional files

## Supplementary files

• Supplementary file 1. siRNA screen identifies BTB domain containing proteins relevant for IAV infection. A549 cells were depleted of individual BTB domain containing proteins by using up to 4 different siRNAs in a 96-well plate setting. After 72 hr of transfection cells were infected with IAV. Infected cells were visualized by immunofluorescence staining of the viral protein NP. The assay was quantified as described in the legend to *Figure 1* and plotted as percentage (%) of NP positive cells compared to control (siControl). BTB domain containing proteins were counted as a hit, if IAV infection was reduced with at least 2 different siRNAs by 49% or more compared to siControl.

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
