## [Decision Letter]

Thank you for submitting your work entitled "A SPOPL/Cullin-3 ubiquitin ligase complex regulates endocytic trafficking by targeting EPS15 at endosomes" for consideration by *eLife*. Your article has been favorably evaluated by Ivan Dikic (Senior editor) and three reviewers, one of whom, Wade Harper, is a member of our Board of Reviewing Editors, and another is Jason MacGurn.

The reviewers have discussed the reviews with one another and the Reviewing Editor has drafted this decision to help you prepare a revised submission.

Summary:

Your paper has now been seen by three reviewers. Overall, the reviewers think the paper is potentially interesting and appropriate for *eLife* but in its current form falls short of what is necessary. In particular, the findings presented do not result in a clear and coherent model for the molecular function of the Cul3-SPOPL E3 ubiquitin ligase with respect to endosomal trafficking. In particular, the observation that SPOPL is required for MVB formation and for IAV entry, but EGFR is trafficked to the lysosome for degradation more rapidly in SPOPL-depleted cells suggests a significant disconnect in terms of the mechanisms proposed. The authors do attempt to address this apparent discrepancy in the Discussion (subsection “EPS15 regulates EGFR trafficking at several steps”, last paragraph) by suggesting that increased interaction between EPS15 and ESCRT-0 in SPOPL-depleted cells may enhance the trafficking of EPS15-interacting cargo (like EGFR) at the expense of other EPS15-independent cargos (presumably IAV). This seems like a reasonable hypothesis – and it is intriguing to speculate that Cul3-SPOPL could regulate cargo flux into the MVB pathway along a specific route – but there are many other observations that complicate this interpretation. For example, the morphological data (Figure 2) and alterations to the abundance of multiple trafficking proteins (EEA1, HRS, STAM, EPS15, LAMP1, AP-2, almost every piece of trafficking machinery they probe in Figure 2 and Figure 3) suggest gross trafficking defects along the endocytic route. This is further complicated by the observation that depletion of SPOPL does not phenocopy depletion of CUL3 with respect to EGFR trafficking – and in fact results in an opposite trafficking phenotype.

So ultimately, the paper needs further work to elucidate a coherent mechanism. After discussions between the reviewers, we feel that the following experiments and improvements would help clarify the underlying mechanisms.

Essential revisions:

1) In the paper, you speculate that increased EPS15 engagement of ESCRT-0 simultaneously promotes EGFR trafficking to the lysosome but abrogates trafficking of cargo that don't engage EPS15. You could test this by knocking down SPOPL in a context where EPS15 cannot engage HRS (either a binding mutant or an EPS15 knockdown), which should restore the trafficking of non-EPS15 cargo. For example, coordinate knockdown of SPOPL and EPS15 should restore IAV infectivity/uncoating. Such an experiment would demonstrate that the observed trafficking defects are indeed dependent on excessive engagement of ESCRT-0 by EPS15.

2) Analysis of the trafficking of additional EPS15-dependent and EPS15-independent cargo would improve your ability to draw conclusions about general vs. specific trafficking defects.

3) The fact that every trafficking component probed exhibits altered stability when SPOPL is depleted is both interesting and problematic – since altered stability/abundance of factors like AP-2 could have severe trafficking consequences. To address the concern that the global membrane trafficking network is grossly perturbed in the absence of SPOPL and CUL3, you may consider probing how loss of Cul3-SPOPL affects trafficking more broadly (clathrin, EPS15R, epsins, other ESCRT components, GGAs, etc.). This would not need to be exhaustive, but it could reveal key insights into just how grossly perturbed trafficking is in these cells.

4) It would improve the paper to gain a sense of what is driving the de-stabilization of these factors in the absence of SPOPL. When SPOPL is depleted, is the decreased stability of other trafficking factors (LAMP1, AP-2) proteasome- or lysosome-dependent? (Presumably this is not mediated by some other CUL3 complex since loss of CUL3 also destabilized AP2.)

5) Some experiments still rely too heavily on overexpressed proteins. Further analysis using endogenous proteins is needed. In Figure 3, it is critical to show that treatment of MG132 or MLN4924 can elevate protein levels of endogenous EPS15. Also demonstrating physical association of EPS15 and SPOPL at endogenous protein levels is needed.

6) An analysis of the EPS15 mutant that cannot be ubiquitylated is needed to further understand the mechanisms involved and to fully demonstrate a role for SPOPL-dependent EPS15 ubiquitylation in the phenotypes observed.

7) Finally, the presentation of the RNAi screen is sub-optimal in that the identities of the other BTB proteins are not being revealed.

---

## [Author Response]

Summary:

*Your paper has now been seen by three reviewers. Overall, the reviewers think the paper is potentially interesting and appropriate for eLife but in its current form falls short of what is necessary. In particular, the findings presented do not result in a clear and coherent model for the molecular function of the Cul3-SPOPL E3 ubiquitin ligase with respect to endosomal trafficking. In particular, the observation that SPOPL is required for MVB formation and for IAV entry, but EGFR is trafficked to the lysosome for degradation more rapidly in SPOPL-depleted cells suggests a significant disconnect in terms of the mechanisms proposed. The authors do attempt to address this apparent discrepancy in the Discussion (subsection “EPS15 regulates EGFR trafficking at several steps”, last paragraph) by suggesting that increased interaction between EPS15 and ESCRT-0 in SPOPL-depleted cells may enhance the trafficking of EPS15-interacting cargo (like EGFR) at the expense of other EPS15-independent cargos (presumably IAV). This seems like a reasonable hypothesis – and it is intriguing to speculate that Cul3-SPOPL could regulate cargo flux into the MVB pathway along a specific route – but there are many other observations that complicate this interpretation. For example, the morphological data (Figure 2) and alterations to the abundance of multiple trafficking proteins (EEA1, HRS, STAM, EPS15, LAMP1, AP-2, almost every piece of trafficking machinery they probe in Figure 2 and Figure 3) suggest gross trafficking defects along the endocytic route. This is further complicated by the observation that depletion of SPOPL does not phenocopy depletion of CUL3 with respect to EGFR trafficking – and in fact results in an opposite trafficking phenotype.*

We agree, the observation that IAV infection is impaired while EGFR degradation is increased upon SPOPL depletion seems puzzling at first. As suggested, we have thus expanded the analysis of trafficking pathways in cells lacking SPOPL. Interestingly, our data now show that SPOPL depletion does not cause gross perturbations of the endosomal system but that SPOPL has rather specific effects on EPS15-dependent cargo and MVB homeostasis (revised Figure 2). IAV infection relies on proper MVB formation, but also on EGFR turnover. In fact, depletion of EGFR alone strongly impairs IAV infection (see Figure 7), and it is thus not surprising that regulation of EGFR trafficking by CRL3^SPOPL^ has a strong impact on IAV infection.

Author response image 1.EGFR depletion inhibits influenza A virus uncoating.**DOI:**
http://dx.doi.org/10.7554/eLife.13841.015

We observe that binding of EPS15 and HRS is increased (Figure 5), while the interaction between EPS15 and EGFR is decreased (Figure 5—figure supplement 1), implying that EPS15 ubiquitination enhances EGFR degradation by altering its binding properties to EGFR. Supporting this hypothesis, it was recently shown that the DUB enzyme USP9X deubiquitinates EPS15 and increases its on/off binding dynamics to EGFR, thereby similarly enhancing EGFR degradation (Savio et al., 2016). Together, these data provide strong support that ubiquitination of EPS15 by CRL3^SPOPL^ and de-ubiquitination by USP9X orchestrate EGFR trafficking to lysosomes at MVBs (Figure 6).

Most of the trafficking defects are shared in cells depleted for CUL3 and SPOPL, with the notable exception of EGFR. Indeed, EGFR levels increase in cells depleted for CUL3 or treated with the CRL-inhibitor MLN-4924, but decrease in cells depleted for SPOPL. This phenotypic difference is however not surprising, since unlike SPOPL, CUL3 regulates several additional steps required for IAV infection. In fact, as shown in the revised Figure 1, we identified many CUL3 adaptor proteins that influence IAV infection and might have an impact on endocytosis. Thus, while SPOPL regulates later steps of cargo delivery to LE / lysosomes, other adaptors might influence cargo uptake or other trafficking steps not affected by SPOPL. Therefore, depletion of CUL3 might inhibit EGFR degradation at other steps, while SPOPL specifically regulates the fate of EGFR at late endosomes/MVB formation. In summary, the observed difference in EGFR degradation in CUL3 and SPOPL depleted cells does not indicate that SPOPL functions independently of CUL3, but rather that CUL3 uses adaptors other than SPOPL to regulate additional steps that block EGFR trafficking to lysosomes (Huotari et al., 2012). Further studies are thus needed to unravel the molecular function of the here listed CUL3 adaptors for endocytosis, EGFR trafficking and IAV infection.

*Essential revisions: 1) In the paper, you speculate that increased EPS15 engagement of ESCRT-0 simultaneously promotes EGFR trafficking to the lysosome but abrogates trafficking of cargo that don't engage EPS15. You could test this by knocking down SPOPL in a context where EPS15 cannot engage HRS (either a binding mutant or an EPS15 knockdown), which should restore the trafficking of non-EPS15 cargo. For example, coordinate knockdown of SPOPL and EPS15 should restore IAV infectivity/uncoating. Such an experiment would demonstrate that the observed trafficking defects are indeed dependent on excessive engagement of ESCRT-0 by EPS15.* As suggested by the reviewers, we performed additional experiments. The result supports the model of an excessive engagement of ESCRT-0 with EPS15 in SPOPL-depleted cells. We could now demonstrate that a simultaneous knockdown of EPS15 and SPOPL partially restores EGFR levels (new Figure 5), supporting the notion that SPOPL regulates EPS15 sorting function at late endosomes/MVB’s. We also tried to rescue IAV infection/uncoating by co-depleting EPS15 and SPOPL. However, EPS15 depletion alone already caused a significant reduction in IAV infection (see Figure 8) indicating a broader function in endocytic traffic. Indeed, upon depletion of EPS15 we observed an increase in the ESCRT-0 and ESCRT-I components HRS, STAM and TSG101 (see Figure 8).

Furthermore, LDL trafficking was impaired and late endosomes were strongly enlarged upon EPS15 depletion (see Figure 8). Together, these data suggest that EPS15 is not only responsible for cargo recognition and delivery to multivesicular bodies but also for the formation of ILV and proper late endosomal biogenesis itself. Not surprisingly therefore, double knockdown of SPOPL and EPS15 does not rescue IAV infection but instead has an additive effect (see Figure 8). Even if HRS and STAM are freed from EPS15 during SPOPL depletion, IAV infection will still be inhibited due to broader changes in endosomal traffic caused by EPS15 depletion. We thus believe that this new data identified EPS15 as a critical factor in not only cargo internalization but also late endosomal progression. This underlines the importance of regulation of endosome maturation by SPOPL.

Author response image 2.EPS15 depletion affects late endosomal maturation.(**A**) Co-depletion of SPOPL an EPS15 has an additive effect on influenza A virus infection. (**B**) EPS15 depletion stabilizes ESCRT components HRS, STAM and TSG101. (**C**) EPS15 depletion affects LDL uptake in cells resulting in an accumulation of LDL in enlarged vacuoles (upper panel). Late endosomes, visualized by life-cell microscopy of GFP-RAB7, are enlarged in cells depleted of EPS15 (lower panel).**DOI:**
http://dx.doi.org/10.7554/eLife.13841.016

*2) Analysis of the trafficking of additional EPS15-dependent and EPS15-independent cargo would improve your ability to draw conclusions about general vs. specific trafficking defects.*

EPS15 plays a dual function for internalization of endocytic cargo. In addition to promoting the formation of clathrin-coated pits via interaction with AP2, it targets ubiquitinated proteins, such as EGFR, for endosomal uptake and lysosomal degradation (Li et al., 2014; Torrisi et al., 1999; van Delft et al., 1997). To evaluate whether SPOPL regulates general or specific effects of EPS15 we analyzed a number of receptor tyrosine kinases, classic targets of clathrin-dependent endocytosis, upon SPOPL depletion via western blotting (new Figure 2). Strikingly, out of five receptors only two were affected by SPOPL depletion: EGFR and MET, both known to associate directly with EPS15 (Parachoniak and Park, 2009; Torrisi et al., 1999). In contrast, the level of VEGFR3, IGF1R and even HER2, a close relative of EGFR, were not altered in SPOPL-depleted cells. These new data thus indicate that SPOPL specifically regulates the turnover of cargo that is directly targeted by EPS15.

*3) The fact that every trafficking component probed exhibits altered stability when SPOPL is depleted is both interesting and problematic – since altered stability/abundance of factors like AP-2 could have severe trafficking consequences. To address the concern that the global membrane trafficking network is grossly perturbed in the absence of SPOPL and CUL3, you may consider probing how loss of Cul3-SPOPL affects trafficking more broadly (clathrin, EPS15R, epsins, other ESCRT components, GGAs, etc.). This would not need to be exhaustive, but it could reveal key insights into just how grossly perturbed trafficking is in these cells.* To evaluate whether SPOPL depletion causes global changes in endosomal trafficking we analyzed additional endocytic trafficking markers in two different cell lines (new Figure 2). Importantly, we did not observe strong defects in early steps of endocytosis (clathrin, caveolin), the recycling pathway (RAB11 level), and ER homeostasis (Calnexin). Even the levels of RAB7, a marker for late endosomes did not change upon SPOPL depletion, despite late endosomes being strongly enlarged (Figure 2). However, we did observe a strong effect on proteins located to endosomes and multivesicular bodies, including the ESCRT-0 components EPS15, HRS and STAM, as well as ESCRT-I component TSG101. In contrast, the ESCRT-III component CHMP6 remained unaffected, suggesting dysregulation of the early ESCRT system involved in cargo-recognition and sorting. Importantly, while EPS15 was stabilized by SPOPL depletion in both cell lines, only minor effects were observed on its close relatives EPS15R and Epsin, underlining the specificity of SPOPL-induced EPS15 ubiquitination. In summary these data show that SPOPL-mediated ubiquitination of EPS15 causes specific defects at the step of ILV formation of multivesicular bodies resulting in dysregulation of late endosomal biogenesis. This new data is now shown in the revised Figure 2, and its implications are discussed in the revised text.

*4) It would improve the paper to gain a sense of what is driving the de-stabilization of these factors in the absence of SPOPL. When SPOPL is depleted, is the decreased stability of other trafficking factors (LAMP1, AP-2) proteasome- or lysosome-dependent? (Presumably this is not mediated by some other CUL3 complex since loss of CUL3 also destabilized AP2.)* Upon further analysis we noticed that LAMP1 protein levels were only decreased by SPOPL depletion in HeLa cells while no such effect could be observed in A549 cells (new Figure 2). Similarly, AP2 reduction was cell line specific, and required efficient SPOPL depletion for rather long time periods (72 hours). It thus seems likely that the destabilizing effect on both proteins caused by SPOPL depletion is of secondary nature and most likely represents a context-dependent maturation defect. Co-treatment with proteasomal or lysosomal inhibitors would therefore be difficult to interpret and most likely cause unspecific changes in protein levels. After careful analysis of additional endosomal markers (see above, new Figure 2), we concluded that SPOPL depletion causes trafficking defects mainly at multivesicular bodies and affects early endosomes and lysosomes only in a secondary manner. We have changed the revised text accordingly.

*5) Some experiments still rely too heavily on overexpressed proteins. Further analysis using endogenous proteins is needed. In Figure 3, it is critical to show that treatment of MG132 or MLN4924 can elevate protein levels of endogenous EPS15. Also demonstrating physical association of EPS15 and SPOPL at endogenous protein levels is needed.* To address this point, we first performed co-immunoprecipitation experiments between endogenous EPS15 and SPOPL. As shown in the revised Figure 3 (new panel D), EPS15 specifically interacts with SPOPL, but not SPOP, supporting the cell fractionation data. Moreover, we also analyzed co-localization of endogenous EPS15 with GFP-tagged SPOPL (new Figure 3). Together, these new data demonstrate that EPS15 specifically interacts with SPOPL at vesicular-like, late endosomal structures, and are now discussed accordingly in the revised result section. Finally, we also analyzed the levels of endogenous EPS15 in cells treated with the proteasome inhibitor MG132 or the CRL inhibitor MLN-4924. While we indeed observe a stabilization of GFP-tagged exogenous EPS15 as well as endogenous EPS15 in MG132-treated cells (Figure 3 and Figure 3—figure supplement 1), we do not observe a convincing stabilization of endogenous EPS15 levels after CRL-inhibition after MLN-4924 treatment (Figure 3—figure supplement 1). While we do not fully understand the reason for this observation, we assume that the pleiotropic effects on CRL inhibition may counteract the expected CUL3-SPOPL-mediated EPS15 stabilization in this situation. This hypothesis is further strengthened by the fact that the levels of ESCRT components are not increased upon MLN-4924 treatment (see Figure 2—figure supplement 1). We have now adjusted the revised text accordingly.

*6) An analysis of the EPS15 mutant that cannot be ubiquitylated is needed to further understand the mechanisms involved and to fully demonstrate a role for SPOPL-dependent EPS15 ubiquitylation in the phenotypes observed.* We found that EPS15 is ubiquitinated in a SPOPL-dependent manner on lysine 793 in vivo. As suggested, we now generated the corresponding non–ubiquitinatable K793R EPS15 mutant, and characterized its cellular defects (new Figure 4). As expected, GFP-tagged EPS15-K793R localized to vesicle-like cytoplasmic structures (Figure 4—figure supplement 1), but its overall steady-state level was slightly lower compared to wild-type controls. Importantly however, the EPS15-K793R level did not increase upon SPOPL depletion (Figure 4), and EPS15-K793R is also not degraded upon overexpression of SPOPL in HeLa cells (Figure 4). In addition to the non-binding EPS15^S744-S746A^ mutant, this analysis strengthens our claim that SPOPL-mediated ubiquitination of lysine 793 contributes to EPS15 destabilization and regulates EPS15 activity towards EGFR sorting.

7) Finally, the presentation of the RNAi screen is sub-optimal in that the identities of the other BTB proteins are not being revealed.

As requested, we have changed Figure 1 and now list all BTB-proteins identified in the IAV infection screen. In addition, we included a summary of the screening in [Supplementary-material SD1-data].